# Origo: Interpretable Multi-physics PDE Foundation Model through Neural Operator Splitting

Li Sun[1]   Hongbo Lv[2]   Zhikai Jiang[2]   Zhongtian Sun[3]   Lanxu Yang[2]   Philip S. Yu[4]

## Abstract

Partial Differential Equations (PDEs) play a fundamental role in scientific computing, and recent efforts have sought to extend the success of foundation models to PDE solving. However, multi-physics PDE pre-training faces the unique challenge of disentangling dynamic heterogeneity to learn universal, elementary patterns that generalize to new PDEs. Additionally, cross-physics transfer lacks a theoretical framework for interpretability—specifically, understanding which pre-trained operator knowledge is effectively transferred to target PDEs. To bridge these gaps, we introduce the theory of neural operator splitting, which decomposes PDE evolution into a modulated global spectral operator and sparse local constitutive mechanisms. A key innovation is Origo, which provides a neural operator bank that enables the identification of operator-level generalization patterns. Extensive experiments demonstrate strong zero-shot generalization and mechanism-level interpretability on unseen PDEs.

## 1. Introduction

Partial Differential Equations (PDEs) are fundamental for modeling physical systems, yet traditional numerical methods are often computationally expensive for complex scenarios. To address these challenges, researchers have developed neural operators like FNO (Li et al., 2020a) and DeepONet (Lu et al., 2021) that map inputs directly to solutions, enabling highly efficient simulation. More recently, inspired by the revolutionary success of foundation models in Natu-

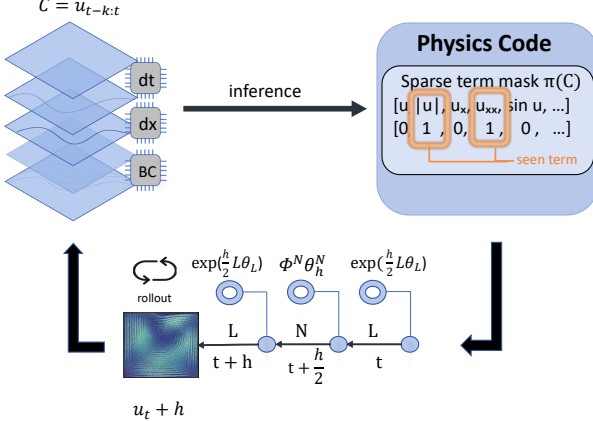

*Figure 1.* Zero-shot generalization on $u_t = |u| + u_{xx}$. Origo successfully predicts the unseen dynamics by recombining learned physical mechanisms.

ral Language Processing (NLP), the focus has shifted from these specialized, single-task models toward a pretrain-and-adapt strategy. Under this paradigm, a unified architecture is trained on diverse physical data to distill universal patterns, thereby establishing a versatile foundation that can be efficiently adapted to unseen PDE problems.

Initial efforts toward such foundation models have largely focused on scaling architectures and leveraging text-based prompts. Representative works include: (i) equation- or text-conditioned frameworks (e.g., ICON-LM (Yang et al., 2023) and PITT (Lorsung et al., 2024)), which align textual captions or tokenized equations with physical states to facilitate generalization to unseen PDEs; (ii) Transformer-based architectures (e.g., FactorFormer (Li et al., 2023)), which enable efficient pretraining on high-resolution spatiotemporal fields by factorizing attention mechanisms, thereby making large-scale multi-domain training feasible; and (iii) large-scale pretraining frameworks (e.g., MPP (McCabe et al., 2023), Poseidon (Herde et al., 2024), and DPOT (Hao et al., 2024)), which train unified backbones on heterogeneous PDE datasets to enhance transferability across different systems.Nevertheless, despite these promising efforts, the field still faces fundamental hurdles regarding negative transfer and physical mechanism entanglement.

[1]Beijing University of Posts and Telecommunications, Beijing, China [2]School of Control and Computer Engineering, North China Electric Power University, Beijing, China [3]Department of Computer Science, University of Kent, Canterbury, United Kingdom [4]Department of Computer Science, University of Illinois Chicago, Chicago, IL, USA. Correspondence to: Li Sun <lsun@bupt.edu.cn>.

*Proceedings of the 43rd International Conference on Machine Learning*, Seoul, South Korea. PMLR 306, 2026. Copyright 2026 by the author(s).

First, negative transfer arises from the intrinsic mathematical disparities among diverse physical systems. Different PDEs often exhibit conflicting dynamic behaviors—such as the smoothness of parabolic diffusion versus the sharp discontinuities of hyperbolic advection (Evans, 2010; Caruana, 1997). Training a single shared backbone on such heterogeneous tasks creates optimization trade-offs, where optimization for one physics regime degrades performance on another.Second, physical mechanism entanglement stems from the monolithic operator modeling prevalent in large-scale PDE pretraining: the update rule is learned as a single densely-coupled function rather than an explicitly structured composition of physical sub-operators (McCabe et al., 2023; Herde et al., 2024). As a result, distinct operators are blended within shared representations, making it difficult to isolate, activate, or manipulate individual physical laws in a controlled manner.

To address these challenges, we draw inspiration from classical operator splitting theory in numerical analysis (Hairer et al., 2006). This framework decomposes the time evolution of complex systems into a sequence of simpler sub-problems, often separating dominant linear operators from nonlinear interactions. From a theoretical perspective, such decomposition mitigates negative transfer by allocating dedicated evolution operators to competing mechanisms, while rendering their coupling explicit and controllable through sequential composition. However, directly applying classical splitting to foundation models is limited by two factors: (1) classical methods assume explicit knowledge of governing equations, whereas foundation models must infer physics directly from data; and (2) standard solvers rely on fixed, hand-designed splitting schemes, lacking the adaptivity required for multi-regime physics.

To bridge this gap, we propose Origo. We generalize the classical splitting principle into a theory of *Neural Operator Splitting*. This framework overcomes the limitations of traditional numerical solvers by transforming static schemes into learnable, context-adaptive architectures.Specifically, we design two synergistic components: the Mechanism Identification Module and the Operator Evolution Module. The Mechanism Identification Module addresses structural rigidity by dynamically inferring the optimal splitting strategy from the input context rather than relying on fixed templates,thereby effectively isolating interference arising from heterogeneous PDE data types. Guided by this strategy, the operator evolution module explicitly separates linear transport from nonlinear interactions by employing a parameterizable Strang splitting scheme, thereby solving the implicit mechanical problem.

Our contributions are summarized as follows:

- We introduce *neural operator splitting*, a principled theory and architecture for disentangling physical processes, which serves as a structured backbone for multi-physics pretraining.

- We propose Origo, which separates *mechanism inference* from *state evolution* by inferring a physics code from a short context window and executing it with neural operator splitting framework.

- Origo further instantiates an explicit *operator bank* with Entmax-sparse routing and KAN-parameterized constitutive laws, improving interpretability and compositional generalization.

## 2. Related Work

**Neural Operators.** Neural operators have emerged as a powerful paradigm for learning resolution-independent solution mappings of PDEs. Seminal works established the theoretical foundations using Green's function approximations (Lu et al., 2021) and spectral convolutions (Li et al., 2020a). Subsequent research has focused on enhancing architectural expressivity and efficiency. Transformer-based variants (Li et al., 2022; Cao, 2021; Holzschuh et al., 2025) leverage attention mechanisms to capture global dependencies, while graph-based approaches (Li et al., 2020b; Brandstetter et al., 2022b) excel at handling irregular meshes. Recent efforts have also explored integrating physical inductive biases, such as symmetry preservation (Helwig et al., 2023) or Lie-group structures (Brandstetter et al., 2022a), or encoding physical differential equations over complex geometries for entropy-increasing dynamics (Sun et al., 2025b), to improve generalization. However, despite these architectural advancements, existing neural operator methods typically necessitate task-specific training with extensive domain-specific data and often suffer from limited generalization capabilities across varying physical settings.

**Pre-training in Scientific Machine Learning.** Inspired by the success of Large Language Models (Brown et al., 2020; Bommasani et al., 2021; Achiam et al., 2023; Sun & Yu, 2026), the field has rapidly shifted towards developing foundation models for PDEs. Early attempts focused on single-physics pre-training, but recent works aim for universal generalist solvers. To handle the heterogeneity of physical systems, researchers have introduced unified architectures capable of processing diverse spatial dimensions (Chen et al., 2024), arbitrary geometries, and multi-component interactions (Zhang et al., 2024). Concurrently, efficiency-oriented models (Herde et al., 2024; Siddik et al., 2025) employ parameter-efficient techniques to scale to high-resolution turbulence, while generative frameworks (Li et al., 2025; Huang et al., 2024) bridge operator learning with probabilistic modeling to capture stochastic dynamics. While recent advances in graph foundation models have demonstrated that embedding structural priors and

Riemannian geometry can successfully align and unify heterogeneous domains (Sun et al., 2025a; 2026b;a), current PDE pre-training largely relies on dense neural architectures that overlook the intrinsic correlations between different PDE datasets, which can constrain model performance. Consequently, the application of foundation models to scientific computing—particularly for general-purpose PDE solving—remains an open frontier for development.

# 3. Theoretical Framework

## 3.1. Problem Definition

We consider parametrized time-dependent PDEs with state variable $u(x,t) \in \mathbb{R}^m$ defined on $(x,t) \in \Omega \times \mathcal{T} \subset \mathbb{R}^{d+1}$:

$$\partial_t u - \mathcal{F}[u;\theta](x,t) = 0, \quad (x,t) \in \Omega \times \mathcal{T}, \quad (1)$$

with initial condition $u(x,0) = u_0(x)$ and boundary condition $\mathcal{B}[u](x,t) = 0$ on $\partial\Omega$. Here $\mathcal{F}[u;\theta]$ denotes a differential operator parameterized by unknown physics parameters $\theta \in \Theta$.

## 3.2. Limitations of Classical Splitting

We focus on semi-linear evolution equations of the form $\partial_t u = \mathcal{L}u + \mathcal{N}(u)$, where $\mathcal{L}$ captures global transport and $\mathcal{N}$ captures local interactions. Classical operator splitting (e.g., Strang splitting) approximates the evolution operator $\Psi_h$ via

$$u(t+h) \approx e^{\frac{h}{2}\mathcal{L}} \circ e^{h\mathcal{N}} \circ e^{\frac{h}{2}\mathcal{L}} u(t), \quad (2)$$

achieving $\mathcal{O}(h^2)$ accuracy when $\mathcal{L}$ and $\mathcal{N}$ are explicitly known.

In foundation-model pretraining settings, two assumptions underlying classical splitting break down. First, $\mathcal{L}$ and $\mathcal{N}$ are not analytically specified because the governing physics parameters are unobservable; the operators must be inferred from data. Second, fixed splitting templates are insufficient for multi-physics pretraining, which requires a composition rule that can adapt across heterogeneous regimes.

To bridge this gap, we introduce two shifts while preserving the Strang structure. We move from explicit to latent by replacing hand-designed operators $(\mathcal{L}, \mathcal{N})$ with learnable surrogates $(\Phi_L, \Phi_N)$ conditioned on a latent physics code $z \in \mathcal{Z}$. We move from static to adaptive by keeping the symmetric composition $\Phi_{h/2}^L \circ \Phi_h^N \circ \Phi_{h/2}^L$ formally fixed, while making its parameterization $z$-dependent and inferable from context. These shifts define our Neural Operator Splitting framework.

## 3.3. Neural Operator Splitting

### 3.3.1. LATENT-CONDITIONED STRANG SPLITTING

Let $\Psi_h$ denote the true evolution operator satisfying $u(t + h) = \Psi_h(u(t))$. We construct a neural approximation $\hat{\Psi}_h(\cdot; z)$ parameterized by a latent physics code $z \in \mathcal{Z}$ inferred from context $C = u_{t-k:t}$, and constrain it to preserve the Strang structure:

$$\hat{\Psi}_h(\cdot; z) = \Phi_{h/2}^L(\cdot; z) \circ \Phi_h^N(\cdot; z) \circ \Phi_{h/2}^L(\cdot; z), \quad (3)$$

where $\Phi_L$ and $\Phi_N$ are surrogates for linear and nonlinear sub-dynamics. The code $z$ is inferred by the Mechanism Inference module and then parameterizes $\Phi_L$ and $\Phi_N$ for evolution, enabling zero-shot generalization within the modeled family by inferring a new $z$ without retraining the operators.

### 3.3.2. SPECTRAL AND CONSTITUTIVE PARAMETERIZATION

The splitting form in Eq. (3) specifies how sub-flows are composed; accuracy also depends on how each sub-operator is parameterized. We use domain-adapted designs aligned with the distinct structure of $\mathcal{L}$ and $\mathcal{N}$.

**Spectral Linear Parameterization.** On domains admitting a spectral decomposition, we parameterize the linear sub-operator as a spectral multiplier:

$$\Phi_h^L(u; z) := \mathcal{F}^{-1}[\Lambda(\xi; z, h) \odot \mathcal{F}(u)], \quad (4)$$

where $\mathcal{F}$ is a geometry-adapted spectral transform (e.g., Fourier for periodic domains and sine/cosine bases for Dirichlet/Neumann boundaries), and $\Lambda(\xi; z, h) \in \mathbb{C}^{m \times m}$ is a matrix-valued multiplier conditioned on $z$ and $h$. This choice targets linear semigroups that are spectrally diagonalizable under the assumed boundary conditions.

**Local Constitutive Parameterization.** We parameterize the nonlinear sub-operator as a local time-stepping map:

$$\Phi_h^N(u; z) := u + h \cdot \mathcal{R}_h(u, \nabla u, \dots; z), \quad (5)$$

where $\mathcal{R}_h(\cdot; z)$ is an $h$-dependent nonlinear functional. The hypothesis class is not restricted to first-order Euler residuals: $\mathcal{R}_h$ can be implemented with deeper residual blocks, neural ODE solvers, or higher-order integrators, allowing arbitrarily accurate approximation of $e^{h\mathcal{N}}$ under mild regularity assumptions (Theorem 3.2).

### 3.3.3. THEORETICAL ANALYSIS: UNIVERSALITY GUARANTEES

Under Assumption 3.1, the spectral linear parameterization is expressive enough to approximate a broad class of strongly continuous linear semigroups that are spectrally

diagonalizable under the prescribed boundary conditions, while the local constitutive parameterization can approximate continuous local nonlinear maps via standard universal approximation results for neural networks. Consequently, for any semi-linear PDE with a spectrally diagonalizable linear semigroup and locally Lipschitz nonlinearity, there exists a configuration $z^* \in \mathcal{Z}$ such that the symmetric composition in Eq. (3) approximates the true evolution operator to arbitrary precision. We formalize this guarantee in Theorem 3.2 and analyze global error propagation in Theorem 3.3.

### 3.4. Theoretical Guarantees

**Assumption 3.1** (Regularity and well-posed split flows). Let $H$ be a Hilbert space and let $K \subset H$ be compact. Assume: (i) The linear operator $\mathcal{L}$ generates a strongly continuous semigroup $\{e^{t\mathcal{L}}\}_{t\geq 0}$ on $H$, and this semigroup is spectrally diagonalizable under the chosen spectral transform $\mathcal{F}$ (as in Eq. (4)) for the boundary conditions considered. (ii) The nonlinear operator $\mathcal{N} : H \to H$ is locally Lipschitz on a neighborhood of $K$, so the nonlinear flow $\{e^{t\mathcal{N}}\}_{t\geq 0}$ is well-defined and continuous on $K$ for $t \in [0, T]$. (iii) The exact flow $\Psi_t$ of $\partial_t u = \mathcal{L}u + \mathcal{N}(u)$ exists on $[0, T]$ and is Lipschitz on $K$ with constant $L_\Psi$. (iv) The commutator/regularity terms required for the classical Strang splitting truncation error are bounded on $K$, yielding a second-order splitting constant $C_S$.

We now establish guarantees for representational power and stability. Let $H$ be the Hilbert space of states, and assume $\mathcal{L}$ generates a strongly continuous semigroup.

**Theorem 3.2** (Universal Approximation of Split Sub-Dynamics). *Under Assumption 3.1, let $\Phi_h^L = e^{h\mathcal{L}}$ and $\Phi_h^N = e^{h\mathcal{N}}$ denote the exact linear and nonlinear sub-operators. For any error tolerance $\epsilon > 0$ and compact set $K \subset H$, there exists a parameter configuration $z^* \in \mathcal{Z}$ such that:*

$$\sup_{u\in K} \|\Phi_h^L(u; z^*) - \Phi_h^L(u)\| < \epsilon, \qquad (6)$$

$$\sup_{u\in K} \|\Phi_h^N(u; z^*) - \Phi_h^N(u)\| < \epsilon. \qquad (7)$$

*Here $z^*$ represents the optimal physics code that configures sub-operators to match the target dynamics.*

**Theorem 3.3** (Global Error Bound and Convergence). *Assume the exact flow $\Psi_h$ is Lipschitz continuous with constant $L_\Psi$. Let the local consistency error of the splitting step be $\delta(h) := \sup_{u\in K} \|\hat{\Psi}_h(u; z^*) - \Psi_h(u)\|$. After $N$ steps $(T = Nh)$ using Eq. (3), the global error is bounded by:*

$$\|u(T) - \hat{u}_N\| \leq C_S h^2 + \frac{e^{L_\Psi T} - 1}{L_\Psi} \cdot \frac{\delta(h)}{h}, \qquad (8)$$

*where $C_S$ depends on regularity and commutator bounds implied by Assumption 3.1. This bound confirms second-order convergence when $\delta(h) = \mathcal{O}(h^3)$.*

Moreover, the same bound applies to the deployed model using an inferred code $\hat{z}$ by replacing $\delta(h)$ with

$$\delta_{\text{deploy}}(h) := \sup_{u\in K} \|\hat{\Psi}_h(u; \hat{z}) - \Psi_h(u)\|$$
$$\leq \delta(h) + \sup_{u\in K} \|\hat{\Psi}_h(u; \hat{z}) - \hat{\Psi}_h(u; z^*)\|. \qquad (9)$$

Proofs are provided in Appendix B.

## 4. The Origo Framework

### 4.1. Overall Pipeline

Origo follows a pretrain to adapt paradigm. During pretraining, we train the mechanism inference module and the Strang splitting evolution backbone end to end on trajectories from multiple PDE families and parameter regimes. We do not provide equation identifiers or symbolic forms. The model infers solver controls from observed dynamics. After pretraining, Origo runs in a zero shot setting. Fine tuning further adapts Origo to a target system with fewer epochs and a smaller learning rate.

Figure 2 summarizes Origo. Origo takes a short context window $\mathcal{C} = u_{t-k:t}$. The mechanism inference module predicts a control vector $\mathbf{z} = (\pi, \lambda, \theta_\mathcal{L}, \theta_\mathcal{N})$. The router weights $\pi$ sparsely choose atomic operators from the library. The scalar $\lambda$ scales the nonlinear update. The codes $(\theta_\mathcal{L}, \theta_\mathcal{N})$ modulate the spectral linear operator and the nonlinear operator library. The evolution backbone then advances the state from $u_t$ to $u_{t+\Delta t}$ using symmetric Strang splitting. Each step applies a linear half-step, a nonlinear full-step, and a linear half-step. The linear stream acts in the spectral domain. The nonlinear stream uses KAN-based atomic operators.

### 4.2. Mechanism Inference

The Mechanism Inference module maps the context window $\mathcal{C}$ to solver controls. It outputs a splitting strategy for the current regime. We use two stages. The first stage encodes the context. The second stage decodes the controls.

**Physics Encoding.** This stage extracts a latent representation of the observed dynamics. We feed the context window $\mathcal{C}$ into a sequence encoder $E_\phi$, such as a Transformer or CNN. The encoder produces features over the window. We pool them into a vector $\mathbf{h} \in \mathbb{R}^d$,

$$\mathbf{h} = \text{Pool}(E_\phi(\mathcal{C})). \qquad (10)$$

$\mathbf{h}$ represents the physics embedding for the current regime. As the window moves forward, the model updates $\mathbf{h}$ and tracks changes in the dynamics.

**Strategy Decoding.** This stage maps $\mathbf{h}$ to solver controls through projection heads. We decode the sparse mechanism

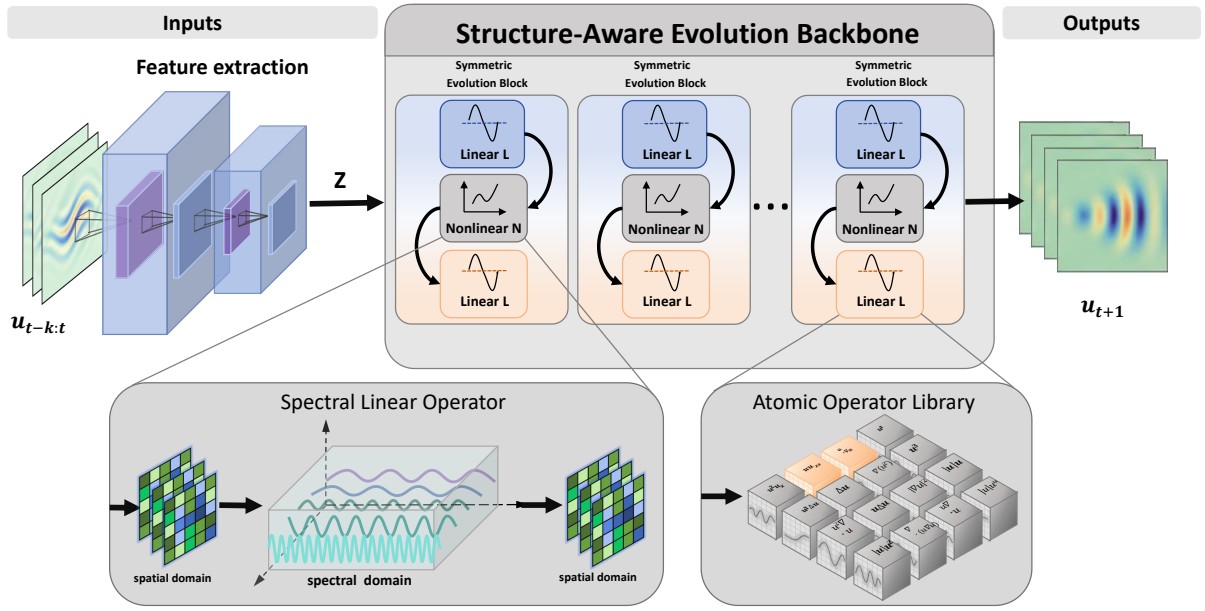

*Figure 2.* **Physics-Encoded Foundation Model.** The model infers latent physical controls from context and evolves the system via a symmetric split-step backbone that combines spectral linear operators with a sparse library of nonlinear atomic operators, enabling structure-aware and transferable PDE rollouts.

selection weights using the $\alpha$-entmax operator:

$$\boldsymbol{\pi} = \text{entmax}_{1.5}(W_\pi \mathbf{h} + b_\pi), \quad (11)$$

where $\pi \in \Delta^{M-1}$ is a sparse probability distribution over the atomic operators. By employing the entmax transformation (with $\alpha = 1.5$), the routing process naturally concentrates on a small, relevant subset of mechanisms by mapping insignificant logits to exact zeros, ensuring a more interpretable and efficient selection.

We also decode the parameters for the linear stream, the nonlinear stream, and the update gate,

$$\theta_\mathcal{L} = W_L \mathbf{h} + b_L, \; \theta_\mathcal{N} = W_N \mathbf{h} + b_N, \; \lambda = \sigma(W_\lambda \mathbf{h} + b_\lambda). \quad (12)$$

$\theta_\mathcal{L}$ and $\theta_\mathcal{N}$ modulate the linear and nonlinear streams. $\lambda$ sets the residual strength. This design conditions the solver configuration on $\mathbf{h}$.

### 4.3. Operator Evolution

The Operator Evolution module updates the state from $u_t$ to $u_{t+\Delta t}$. It uses the inferred controls

$$\mathbf{z} = (\pi, \lambda, \theta_\mathcal{L}, \theta_\mathcal{N}, \rho). \quad (13)$$

The update follows the symmetric Strang splitting rule in Eq. (3). The model keeps the splitting skeleton fixed. The controls in $\mathbf{z}$ adapt the linear stream, the nonlinear stream, and the residual correction.

#### 4.3.1. THE SYMMETRIC SPLITTING SKELETON

Eq. (3) defines one step as three sub-steps. We implement the step as

$$
\begin{aligned}
u^{(1)} &= \Phi_\mathcal{L}^{\Delta t/2}(u_t; \theta_\mathcal{L}), \\
u^{(2)} &= \Phi_\mathcal{N}^{\Delta t}\left(u^{(1)}; \pi, \lambda, \theta_\mathcal{N}\right), \\
u_{t+\Delta t} &= \Phi_\mathcal{L}^{\Delta t/2}\left(u^{(2)}; \theta_\mathcal{L}\right) + \rho \, \mathcal{R}\left(u^{(2)}\right).
\end{aligned}
\quad (14)
$$

$\Phi_\mathcal{L}$ models the linear flow. $\Phi_\mathcal{N}$ models the nonlinear flow. $\mathcal{R}$ adds a residual correction. This order matches Strang splitting.

#### 4.3.2. SPECTRAL LINEAR OPERATOR

The linear stream models global effects such as diffusion and dispersion. Many linear PDE operators act diagonally in frequency space. We use this property and define a diagonal spectral flow:

$$\Phi_\mathcal{L}^\tau(u; \theta_\mathcal{L}) = \mathcal{F}^{-1}\left(\exp\left(\tau \cdot \Lambda(\theta_\mathcal{L})\right) \odot \mathcal{F}(u)\right), \quad (15)$$

where $\mathcal{F}$ is the Fourier transform and $\odot$ denotes element-wise multiplication. $\Lambda(\theta_\mathcal{L}) \in \mathbb{C}^K$ stores mode-wise complex multipliers. The latent code $\theta_\mathcal{L}$ controls these multipliers. This form gives a global receptive field and preserves resolution invariance.

#### 4.3.3. KAN-BASED NONLINEAR OPERATOR

The nonlinear stream models local nonlinear dynamics. We use an atomic operator library and sparse routing. The

routing weights $\pi$ select a small set of active mechanisms. Each mechanism uses a KAN parameterization.

**Atomic Operator Library.** Let $\{\mathcal{O}_m\}_{m=1}^M$ denote candidate nonlinear mechanisms. The nonlinear flow applies

$$\Phi_{\mathcal{N}}^\tau(u; \pi, \lambda, \theta_{\mathcal{N}}) = u + \tau\,\lambda \sum_{m=1}^M \pi_m\,\mathcal{O}_m(u; \theta_{\mathcal{N},m})\,. \quad (16)$$

$\pi_m$ gives the selection weight for the $m$-th mechanism. $\lambda$ sets the overall nonlinear update strength. $\theta_{\mathcal{N},m}$ controls the parameters inside $\mathcal{O}_m$.

**KAN Parameterization.** Each atomic operator uses a KAN to model its constitutive function. We adopt KANs because spline-based univariate components provide strong approximation power for smooth nonlinear laws, while remaining more interpretable than dense MLP features. The KAN uses learnable spline functions. We write the $m$-th constitutive mapping as

$$\Phi_m(u) = \sum_{j=1}^K \phi_{m,j}(u), \quad (17)$$

where $\phi_{m,j}$ are B-spline components with parameters in $\theta_{\mathcal{N},m}$. This design yields two practical benefits: (i) it can fit diverse nonlinear response curves with limited parameters and stable optimization, and (ii) under sparse routing, each activated term can be inspected via its learned spline shape, making the inferred mechanisms directly readable rather than implicit in a black-box network.

### 4.3.4. MODEL BIAS COMPENSATION

The splitting backbone does not capture all discretization errors and all operator interactions. We add a gated residual correction:

$$u_{\text{corr}} = u_{\text{split}} + \rho \cdot \text{CNN}_{\text{res}}(u_{\text{split}})\,, \quad (18)$$

where $\rho \in [0, 1)$ is a scalar gate inferred by the Mechanism Inference module. When $\rho$ approaches zero, the model reduces to the pure splitting solver. The residual term compensates higher-order splitting errors. These errors relate to commutator terms such as $[\mathcal{L}, [\mathcal{L}, \mathcal{N}]]$ and $[\mathcal{N}, [\mathcal{N}, \mathcal{L}]]$.

### 4.4. Training and Adaptation

Origo follows a unified *pretrain-to-adapt* paradigm. We train all components end-to-end during pretraining on heterogeneous PDE trajectories, then run zero-shot or adapt the model to a target system via fine-tuning.

**Long-horizon rollout loss.** A core requirement for PDE surrogates is accurate *multi-step* forecasting, since small

---

**Algorithm 1** End-to-End Training and Inference of Origo

1: **Input:** Context window $\mathcal{C} = u_{t-k:t}$, target trajectory $u_{t+1:t+T}$, time step $\Delta t$, rollout horizon $T$.
2: **Initialize:** Model parameters (encoder $E_\phi$, projection heads, spectral and KAN parameters).
3: % *Phase 1: Mechanism Inference*
4: Extract physics embedding: $h = \text{Pool}(E_\phi(\mathcal{C}))$
5: Decode routing probabilities: $\pi = \text{entmax}_{1.5}(W_\pi h + b_\pi)$
6: Decode operator controls: $\theta_{\mathcal{L}}, \theta_{\mathcal{N}}, \lambda, \rho$ (Eq. 12)
7: % *Phase 2: Operator Evolution*
8: Set initial rollout state: $\hat{u}_t = u_t$
9: **for** $\tau = 1, \dots, T$ **do**
10:    % *Linear half-step*
11:    $u^{(1)} = \mathcal{F}^{-1}\left(\exp\left(\frac{\Delta t}{2}\Lambda(\theta_{\mathcal{L}})\right) \odot \mathcal{F}(\hat{u}_{t+\tau-1})\right)$
12:    % *Nonlinear full-step*
13:    $u^{(2)} = u^{(1)} + \Delta t\lambda \sum_{m=1}^M \pi_m \mathcal{O}_m(u^{(1)}; \theta_{\mathcal{N},m})$
14:    % *Linear half-step*
15:    $u_{split} = \mathcal{F}^{-1}\left(\exp\left(\frac{\Delta t}{2}\Lambda(\theta_{\mathcal{L}})\right) \odot \mathcal{F}(u^{(2)})\right)$
16:    % *Residual Correction*
17:    $\hat{u}_{t+\tau} = u_{split} + \rho \cdot \text{CNN}_{res}(u_{split})$
18: **end for**
19: % *Phase 3: Loss Computation (Pre-training / Fine-tuning)*
20: **if** Training **then**
21:    Compute rollout loss $\mathcal{L}_{roll}$ between $\{\hat{u}_{t+1}, \dots, \hat{u}_{t+T}\}$ and $u_{t+1:t+T}$ (Eq. 19)
22:    Compute sparsity loss $\mathcal{L}_{sparse}$ (Eq. 20) and regularization $\mathcal{L}_{reg}$ (Eq. 21)
23:    $\mathcal{L}_{total} = \mathcal{L}_{roll} + \gamma_s \mathcal{L}_{sparse} + \gamma_r \mathcal{L}_{reg}$
24:    Update model parameters via gradient descent on $\mathcal{L}_{total}$
25: **end if**
26: **Output:** Predicted rollout trajectory $\{\hat{u}_{t+1}, \dots, \hat{u}_{t+T}\}$

---

one-step errors can accumulate and destabilize rollouts. Given a context window $\mathcal{C} = u_{t-k:t}$, Origo infers controls $\mathbf{z}$ and rolls out the evolution module for $T$ steps. We therefore train with a long-horizon objective:

$$\mathcal{L}_{\text{roll}} = \frac{1}{T} \sum_{\tau=1}^T \|\hat{u}_{t+\tau} - u_{t+\tau}\|_2^2\,. \quad (19)$$

This directly aligns optimization with deployment, where the model is used autoregressively over long horizons.

**Sparse routing.** To encourage selective mechanism activation (and avoid dense mixtures), we compute routing weights $\pi$ with entmax and minimize entropy over the rollout horizon:

$$\mathcal{L}_{\text{sparse}} = \frac{1}{T} \sum_{\tau=1}^T \left(-\sum_{m=1}^M \pi_m^{(t+\tau)} \log \pi_m^{(t+\tau)}\right)\,. \quad (20)$$

**Residual regularization.** We constrain the residual gate $\rho$ so that correction remains limited rather than dominating the dynamics:

$$\mathcal{L}_{\text{reg}} = \frac{1}{T} \sum_{\tau=1}^{T} \left\| \rho^{(t+\tau)} \right\|_2^2. \tag{21}$$

**Overall objective.** We optimize the composite loss:

$$\mathcal{L}_{\text{total}} = \mathcal{L}_{\text{roll}} + \gamma_s \mathcal{L}_{\text{sparse}} + \gamma_r \mathcal{L}_{\text{reg}}. \tag{22}$$

**Zero-shot inference and fine-tuning.** After pretraining, Origo runs zero-shot on unseen systems by inferring **z** from context and rolling out without task-specific updates. If target data is available, we fine-tune with the same objective using fewer epochs and a smaller learning rate, while preserving the Strang split-step structure.

### 4.5. Algorithm Summary

Algorithm 1 summarizes Origo's end-to-end training and inference. Given a short context window of past trajectories, the Mechanism Inference module extracts latent physics representations and solver controls. The Operator Evolution module then auto-regressively predicts future states over a horizon $T$. Each step advances via a symmetric Strang splitting skeleton, alternating between a spectral linear global operator and a sparse KAN-based local nonlinear operator. Finally, pre-training jointly minimizes rollout, sparsity, and residual losses, while zero-shot inference executes direct forward rollouts without parameter updates.

## 5. Experiments

This section is organized as follows: we first detail the experimental setup, followed by a comprehensive evaluation of our model's capacity to learn multiple PDEs and generalize to diverse downstream tasks. Finally, we provide an ablation study to analyze the impact of key hyperparameters.

### 5.1. Experimental Setup

**Datasets.** We evaluate Origo on a multi-scale benchmark suite curated from four data sources: FNO (Li et al., 2020a), PDEBench (Takamoto et al., 2022), PDEArena (Gupta & Brandstetter, 2022), and CFDBench (Luo et al., 2023). The suite consists of 10 datasets spanning a wide range of PDE families and parameter regimes, including Navier–Stokes, diffusion–reaction, and shallow-water systems.

**Baselines.** The baselines are divided into two categories: (1) Task-specific expert models, which include U-Net (Ronneberger et al., 2015), FNO (Li et al., 2020a), FFNO (Tran et al., 2021), GK-T (Cao, 2021), and GNOT (Hao et al., 2023). All of these approaches typically require training

from scratch for each specific case. (2) Unified pre-training models, which include DPOT (Hao et al., 2024), MOE-POT (Wang et al., 2025), and OmniArch (Chen et al., 2024). These models are designed to learn generalized representations across diverse physical systems. More details on the baselines are provided in Appendix D.

**Implementation Details.** For all model scales, we utilized the AdamW optimizer with an initial learning rate of $1 \times 10^{-3}$. The models were trained for 1,000 epochs. All experiments were conducted on a computational platform equipped with NVIDIA RTX 5090 GPUs. Consistent with prior works (Li et al., 2020a), we evaluate the prediction accuracy using the relative $L_2$ error (L2RE). Additional training details, including parameter counts and data dimensions, are provided in Appendix D.

### 5.2. Main Results

Table 1 summarizes L2 relative error on multiple PDE benchmarks, with methods grouped into task-specific expert models, unified pre-training, and fine-tuned variants. The first block is competitive on certain settings, but lacks unified transfer across heterogeneous PDE families and regimes, often necessitating per-equation retraining. The second block reports zero-shot results from unified pre-trained backbones, demonstrating strong cross-benchmark generalization without relying on equation IDs. The last block shows fine-tuning for 200 epochs, which improves performance across benchmarks; the fine-tuned unified models achieve the best or near-best accuracy, with Origo-200 delivering the strongest or near-strongest L2RE across most tasks/columns. These results strongly support unified pre-training and adaptation as a scalable route to reusable PDE foundation backbones, outperforming expert-style per-task training in both transferability and final accuracy.

### 5.3. Zero-shot Performance

We evaluated the model's zero-shot generalization capability. The model was pre-trained on a diverse mixture of PDE datasets, with the test equations strictly excluded from the training set. During inference, given a short context window $\mathcal{C} = u_{t-k:t}$, we applied the pre-trained model directly to the target equations without any fine-tuning.

We report results on two unseen time-dependent PDEs: 1D KdV and 2D Allen–Cahn. Although these equations are never observed during training, their constituent operator types (e.g., advection/dispersion for KdV and diffusion/reaction for Allen–Cahn) are covered by the mixed training set. This setting tests compositional generalization: the model must infer a new equation by recombining mechanisms learned from other PDE families.

As shown in Table 2, Origo-M significantly outperforms all

*Table 1.* The L2RE on multiple PDE benchmarks. Methods are grouped into task-specific expert models, unified pre-training, and fine-tuned variants. The previous state-of-the-art results are underlined and our best results are bolded (lower is better).

| L2RE | Params | FNO-$\nu$ | | | PDEBench | | | | PDEArena | | CFDBench |
|---|---|---|---|---|---|---|---|---|---|---|---|
| | | 1e-5 | 1e-4 | 1e-3 | Bur | DR | SWE | CNS-(1,0.1) | NS | NS-cond | — |
| *Task specific Expert Models* | | | | | | | | | | | |
| FNO | 0.5M | 0.182 | 0.0976 | 0.0151 | 0.0205 | 0.141 | 0.00521 | 0.115 | 0.106 | 0.374 | 0.00884 |
| UNet | 25M | 0.232 | 0.140 | 0.0290 | 0.374 | 0.114 | 0.0612 | 0.392 | 0.120 | 0.394 | 0.245 |
| FFNO | 1.3M | 0.142 | 0.0589 | 0.0116 | 0.0141 | 0.0669 | 0.0136 | 0.0248 | 0.0985 | 0.704 | 0.00835 |
| GK-T | 1.6M | 0.158 | 0.0931 | 0.0115 | 0.0147 | 0.0421 | 0.00806 | 0.0400 | 0.111 | 0.494 | 0.0123 |
| GNOT | 1.8M | 0.184 | 0.0518 | 0.0146 | 0.0123 | 0.0365 | 0.00793 | 0.0379 | 0.201 | 0.379 | 0.0103 |
| *Unified Pre-training* | | | | | | | | | | | |
| DPOT-M | 122M | 0.0482 | 0.0334 | 0.00556 | 0.0123 | 0.0342 | 0.00338 | 0.0136 | 0.0950 | 0.323 | 0.00875 |
| MoE-POT-M | 288M | 0.0420 | 0.0410 | 0.00667 | 0.0111 | 0.0350 | 0.00353 | 0.0107 | 0.115 | 0.410 | 0.00599 |
| OmniArch-M | 144M | 0.241 | 0.169 | 0.0530 | 0.0104 | 0.0186 | 0.00188 | 0.0230 | 0.202 | 0.643 | 0.609 |
| Origo-M(Ours) | 97M | 0.0342 | 0.0257 | 0.00483 | 0.00989 | 0.00247 | 0.00327 | 0.00887 | 0.0871 | 0.312 | 0.0125 |
| *Fine-tuned* | | | | | | | | | | | |
| DPOT-200 | 122M | 0.0300 | 0.0168 | 0.00501 | 0.00445 | 0.0166 | 0.00257 | 0.0144 | 0.0384 | 0.225 | 0.00528 |
| MoE-POT-200 | 288M | 0.0410 | 0.0188 | 0.00455 | 0.00410 | 0.0163 | 0.00228 | 0.00870 | 0.0721 | 0.259 | 0.00468 |
| OmniArch-200 | 144M | 0.0963 | 0.0597 | 0.0176 | 0.00831 | 0.0120 | 0.00177 | 0.0179 | 0.112 | 0.444 | 0.00472 |
| Origo-200(Ours) | 97M | **0.0268** | **0.0155** | **0.00437** | **0.00372** | 0.0142 | **0.00153** | **0.00853** | **0.0247** | **0.215** | **0.00443** |

*Table 2.* Zero-shot generalization performance (Relative $L_2$ Error) on unseen 1D KdV and 2D Allen–Cahn equations.

| Method | 1D KdV | 2D Allen–Cahn |
|---|---|---|
| | Rel-$L_2$ $\downarrow$ | Rel-$L_2$ $\downarrow$ |
| DPOT-M | 0.0385 | 0.271 |
| MOE-POT-M | 0.0237 | 0.152 |
| OmniArch-M | 0.0342 | 0.249 |
| **Origo-M (ours)** | 0.0103 | 0.0952 |

baseline methods (DPOT-M, MOE-POT-M, and OmniArch-M) in this challenging setting. specifically, our model achieves the lowest Relative $L_2$ error of 0.0103 on the 1D KdV equation and 0.0952 on the 2D Allen–Cahn equation. Notably, Origo-M reduces the error by approximately 50% to 60% compared to the second-best performing method (MOE-POT-M), demonstrating superior robustness in transferring learned physical priors to unseen dynamics.

## 5.4. Interpretable Experiment

To demonstrate interpretability, we analyze the mechanism representation inferred by Origo on the 1D viscous Burgers' equation, governed by $u_t + (u^2/2)_x = (\nu/\pi)u_{xx}$ with $\nu = 0.1$. By expanding the advection term, the target dynamics can be expressed as $u_t = -1.0 \cdot (uu_x) + 0.0318 \cdot u_{xx}$. We extract context windows from test trajectories and visualize the predicted mechanism weights to inspect the activated physical terms.

The resulting heatmap (Figure 3) confirms that the model accurately uncovers the governing physics with high sparsity. Specifically, the model identifies the nonlinear advection term (interaction between $u$ and $\partial_x$) with a weight of

$-0.9752$, closely approximating the theoretical coefficient of $-1.0$. Furthermore, the diffusion term ($\partial_{xx}$) is recovered with a weight of $0.0311$, which matches the ground truth viscosity coefficient ($\nu/\pi \approx 0.0318$) with a relative error of only $2.2\%$. Crucially, weights for extraneous terms remain near zero, demonstrating that the model successfully isolates the correct symbolic structure without overfitting to irrelevant operators.

To further demonstrate the robustness of this recovery across more complex systems, we extend our interpretability experiments to the Navier-Stokes equations. As shown in Table 3, Origo successfully identifies the dominant physical mechanisms with high precision. The model accurately recovers the coefficient for the convection term ($u \cdot \nabla u$) with a relative error of only $1.8\%$, and the viscosity term ($\nu \nabla^2 u$) with a $5.0\%$ error. Consistent with our observations on the Burgers' equation, the learned weights for irrelevant terms (such as $u^2$ and $|\nabla u|^2$) are rigorously suppressed to near zero, reaffirming the model's capability to correctly isolate the true governing physics.

*Table 3.* Symbolic Term Weight Recovery for Navier-Stokes Equations

| Term | GT | Learned | Rel. Err. |
|---|---|---|---|
| Convection ($u \cdot \nabla u$) | 1.000 | 0.982 | 1.8% |
| Viscosity ($\nu \nabla^2 u$) | 0.001 | 0.00095 | 5.0% |
| Term 1 ($u^2$) | 0.000 | 0.001 | - |
| Term 2 ($|\nabla u|^2$) | 0.000 | 0.000 | - |

*Note:* **GT**: Ground Truth; **Learned**: Learned Weight; **Rel. Err.**: Relative Error. Term 1 and 2 represent irrelevant physical terms.

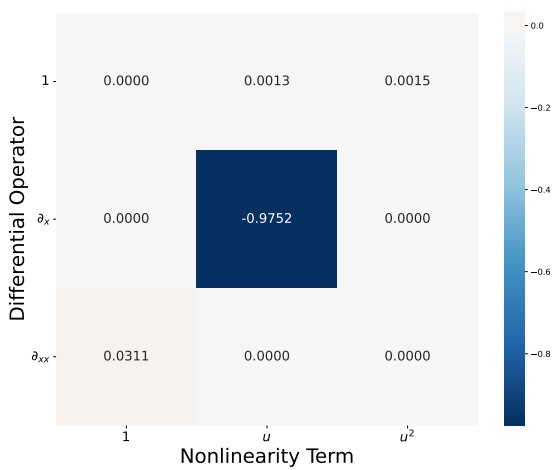

*Figure 3.* Interpretable Mechanism Readout via Sparse Routing.

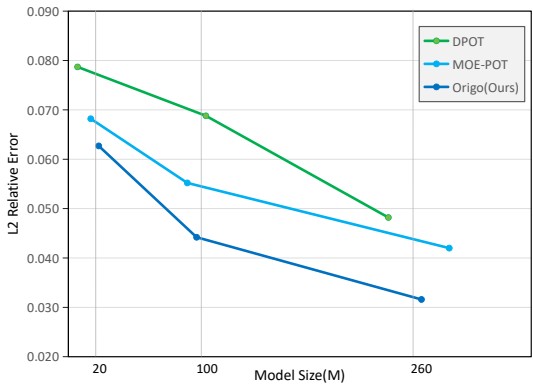

*Figure 4.* Scaling analysis results.

### 5.5. Scaling Experiments

We evaluate the scalability of Origo in Figure 4. The results demonstrate a clear scaling law: as the model size increases from 20M to 260M, the zero-shot L2 Relative Error decreases significantly from 0.063 to 0.032. Origo consistently outperforms both DPOT and MOE-POT across all scales. Notably, our 100M model achieves a lower error ($\sim 0.044$) than the largest 260M DPOT baseline, verifying the superior parameter efficiency of our approach.

*Table 4.* Ablation study on architectural components. Origo (Full) consistently achieves the best performance and effectively prevents negative transfer in zero-shot regimes.

|                  | Standard Tasks |        | Zero-Shot Tasks |               |
|------------------|----------------|--------|-----------------|---------------|
| **Variant**      | **Burgers**    | **NS** | **1D KdV**      | **2D Allen-Cahn** |
| w/o Split-Stepper | 0.0358        | 0.267  | 0.0421          | 0.285         |
| w/o Mechanism     | 0.0231        | 0.143  | 0.0284          | 0.193         |
| **Origo (Full)**  | **0.00989**   | **0.0871** | **0.0103**  | **0.0952**    |

### 5.6. Ablation Studies

To verify our design choices and demonstrate the necessity of our components for preventing negative transfer in unseen regimes, we compare Origo against two variants to isolate the effects of the split-stepper and mechanism inference. Results across both standard and zero-shot tasks are summarized in Table 4.

**Effect of the Split-Stepper.** The w/o Split-Stepper variant replaces the symmetric splitting structure with a standard residual backbone. This leads to the most severe degradation on standard tasks, with relative error on the NS equation increasing drastically ($0.0871 \rightarrow 0.267$). Crucially, this vulnerability extends directly to unseen regimes, where errors surge for the 1D KdV ($0.0103 \rightarrow 0.0421$) and 2D Allen-

Cahn ($0.0952 \rightarrow 0.285$) zero-shot tasks. This confirms that the split-step evolution provides a critical inductive bias, essential for stabilizing rollout dynamics and mitigating error accumulation over long horizons, regardless of the equation type.

**Effect of Mechanism Inference.** The w/o Mechanism Inference variant disables sparse routing and assigns uniform weights to all operators. This results in a significant performance drop on the training distribution (Burgers: $0.00989 \rightarrow 0.0231$; NS: $0.0871 \rightarrow 0.143$), indicating that simply providing an operator library is insufficient. More importantly, in zero-shot scenarios, the absence of this module leads to severe negative transfer, increasing the errors on unseen KdV ($0.0103 \rightarrow 0.0284$) and Allen-Cahn ($0.0952 \rightarrow 0.193$) equations by a factor of two to three. This provides strong empirical evidence that explicit mechanism selection is crucial to avoid operator entanglement and ensure robust generalization to unseen equations.

## 6. Conclusion

In this work, we proposed Origo, a physically interpretable foundation model rooted in Neural Operator Splitting. By disentangling complex dynamics into global spectral operators and local constitutive mechanisms, Origo achieves zero-shot generalization on unseen physical systems while demonstrating superior parameter efficiency. Crucially, our model goes beyond black-box prediction and maintains a structure-aware evolution backbone, which yields glass-box interpretability and recovers underlying physical laws from data. Origo also supports adaptation with limited target trajectories, since the inferred physics code provides a strong initialization for downstream fine-tuning. This work represents a significant step towards trustworthy and explainable foundation models for scientific discovery.

## Acknowledgement

This work is supported in part by NSFC under grant 62202164. Philip S. Yu is supported in part by NSF under grants III-2106758 and POSE-2346158.

## Impact Statement

Origo contributes to the development of interpretable foundation models for scientific computing. By introducing neural operator splitting, it decomposes multi-physics PDE dynamics into global spectral evolution and local constitutive mechanisms, enabling more transparent and transferable PDE solving. This design may benefit scientific and engineering applications where efficient simulation is important, such as fluid dynamics, climate modeling, materials science, and complex physical system analysis. Its ability to generalize to unseen PDEs and expose mechanism-level routing patterns could reduce the cost of adapting neural solvers to new physical regimes while improving trust in model predictions. While Origo is intended to support beneficial scientific and engineering applications, we encourage careful validation before applying learned PDE solvers in safety-critical physical systems.

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

# A. Table of Notations

We summarize key notations used in Section 3 and Appendices B–C in Table A.1.

*Table A.1.* Summary of notations.

| Symbol | Description |
|---|---|
| **Problem setting and data** | |
| $D \subset \mathbb{R}^d$ | Spatial domain of dimension $d$ |
| $t,\ T$ | Continuous time and rollout horizon |
| $h$ | Time step size (discretization), $N = T/h$ steps |
| $u(x, t)$ | Solution field at location $x$ and time $t$ |
| $u_{t-k:t}$ | Observed context window (snapshots) used for inference |
| $\mu$ | PDE/physical parameters (e.g., viscosity, forcing coefficients) |
| **Dynamics and flows (Section 3)** | |
| $L$ | Linear operator (possibly nonlocal), e.g., diffusion/dispersion |
| $N(\cdot)$ | Nonlinear operator (local/pointwise constitutive dynamics) |
| $\Psi_t$ | Exact flow map of the full dynamics for time $t$ |
| $\Phi_h^L = e^{hL}$ | Exact linear sub-flow over step $h$ |
| $\Phi_h^N = e^{hN}$ | Exact nonlinear sub-flow over step $h$ |
| $\Psi_h^S$ | Exact Strang splitting step: $\Phi_{h/2}^L \circ \Phi_h^N \circ \Phi_{h/2}^L$ |
| $C_{\mathrm{disc}}$ | Strang local truncation constant: $\|\Psi_h^S - \Psi_h\| = O(h^3)$ |
| $L_\Psi$ | Lipschitz/stability constant of $\Psi_t$ on $K$ (Assumption 3.1) |
| **Origo / learned splitting model** | |
| $z$ | Latent physics/mechanism code inferred from context |
| $\widehat{\Phi}_h^L(\cdot; z)$ | Learned linear sub-step (spectral multiplier parameterization) |
| $\widehat{\Phi}_h^N(\cdot; z)$ | Learned nonlinear sub-step (local operator bank) |
| $\widehat{\Psi}_h(\cdot; z)$ | Learned Strang step: $\widehat{\Phi}_{h/2}^L \circ \widehat{\Phi}_h^N \circ \widehat{\Phi}_{h/2}^L$ |
| $\widehat{u}_n$ | Model rollout state at step $n$ |
| $u_n$ | Ground-truth state at $t_n = nh$ |
| $e_n = \widehat{u}_n - u_n$ | Global rollout error at step $n$ |
| $\delta(h)$ | One-step consistency error: $\sup_{u \in K} \|\widehat{\Psi}_h(u; z) - \Psi_h(u)\|$ |
| $\widehat{z}$ | Deployed (inferred) code; may differ from the optimal $z^*$ |
| **Spectral transform and operator bank** | |
| $\mathcal{F},\ \mathcal{F}^{-1}$ | Spectral transform (e.g., Fourier/sine/cosine) and inverse |
| $\Lambda(\xi; z, h)$ | Spectral multiplier for the learned linear step (mode $\xi$) |
| $E_i(\cdot)$ | $i$-th nonlinear expert/operator in the local bank (e.g., KAN expert) |
| $\mathbf{w}$ | Router weights over experts (e.g., Entmax outputs) |
| $\varphi$ | Basis functions in spline/KAN parameterization |
| **Spaces and metrics** | |
| $\mathcal{H}$ | Hilbert space for analysis (Appendix B/C) |
| $K \subset \mathcal{H}$ | Compact set where flows are well-defined/stable |
| $H^s(D)$ | Sobolev space of order $s$ on $D$ |
| $\|\cdot\|$ | Norm on $\mathcal{H}$ (context-dependent) |
| $\|\cdot\|_{L^2}$ | $L^2$ norm over $D$ |
| Rel-$L^2$ / nRMSE | Evaluation metrics for rollout accuracy |

# B. Theoretical Proofs

This appendix provides proofs for the theoretical results in Section 4, specifically Theorem 3.2 and Theorem 3.3. Throughout, we follow the standing regularity conditions in Assumption 3.1 and make all constants independent of the step size $h$ and rollout length $N = T/h$.

## B.1. Standing assumptions and notation

Let $\mathcal{H}$ be a Hilbert space with norm $\|\cdot\|$, and let $K \subset \mathcal{H}$ be a compact set. We consider the semi-linear evolution equation

$$\partial_t u = Lu + N(u), \qquad u(0) = u_0 \in \mathcal{H}, \tag{B.1}$$

with exact flow map $\Psi_t : \mathcal{H} \to \mathcal{H}$ so that $u(t) = \Psi_t(u_0)$.

We restate Assumption 3.1 (used in Section 3.4) for convenience.

**Assumption B.1** (Regularity and well-posed split flows (Assumption 3.1)). Assume:

1. **Linear semigroup.** $L$ generates a strongly continuous semigroup $\{e^{tL}\}_{t\geq 0}$ on $\mathcal{H}$. Moreover, under the chosen spectral transform $\mathcal{F}$ (Eq. (4)), the semigroup is spectrally diagonalizable for the boundary conditions considered.

2. **Nonlinear flow.** $N : \mathcal{H} \to \mathcal{H}$ is locally Lipschitz on a neighborhood of $K$, so $\{e^{tN}\}_{t\geq 0}$ exists and is continuous on $K$ for $t \in [0, T]$.

3. **Exact-flow stability.** The exact flow $\Psi_t$ is Lipschitz on $K$ for $t \in [0, T]$ with constant $L_\Psi$, i.e., $\|\Psi_t(u) - \Psi_t(v)\| \leq e^{L_\Psi t}\|u - v\|$ for all $u, v \in K$.

4. **Strang regularity.** The commutator/regularity terms required for the classical Strang splitting truncation error are bounded on $K$, yielding a second-order splitting constant $C_S$.

Define the exact sub-flows for step size $h > 0$:

$$\Phi_h^L := e^{hL}, \qquad \Phi_h^N := e^{hN}.$$

The exact Strang splitting step is

$$\Psi_h^S := \Phi_{h/2}^L \circ \Phi_h^N \circ \Phi_{h/2}^L. \tag{B.2}$$

Our learned split step is the latent-conditioned composition (Eq. (3)):

$$\widehat{\Psi}_h(\cdot; z) := \widehat{\Phi}_{h/2}^L(\cdot; z) \circ \widehat{\Phi}_h^N(\cdot; z) \circ \widehat{\Phi}_{h/2}^L(\cdot; z), \tag{B.3}$$

where $\widehat{\Phi}_h^L$ uses the spectral multiplier parameterization (Eq. (4)) and $\widehat{\Phi}_h^N$ uses a local constitutive map (Eq. (5)).

## B.2. Proof of Theorem 3.2 (Universal approximation of split sub-dynamics)

*Proof.* We prove the two approximation statements (Eq. (6) and Eq. (7)) separately.

**(i) Linear sub-operator.** Under Assumption B.1(1), $\Phi_h^L = e^{hL}$ is spectrally diagonalizable under $\mathcal{F}$ for the boundary conditions of interest. Hence there exists a (matrix-valued) multiplier $\Lambda(\xi; h)$ such that, on the spectral domain,

$$\Phi_h^L(u) = \mathcal{F}^{-1}[\Lambda(\xi; h) \odot \mathcal{F}(u)].$$

Our model class parameterizes $\widehat{\Phi}_h^L(\cdot; z)$ by a learnable multiplier $\Lambda(\xi; z, h)$ (Eq. (4)). Because $K$ is compact and $\mathcal{F}$ is continuous on the discretized/represented domain, for any $\varepsilon > 0$ there exists a code $z^* \in \mathcal{Z}$ such that the induced multiplier $\Lambda(\cdot; z^*, h)$ approximates $\Lambda(\cdot; h)$ uniformly over the finitely represented modes relevant to $K$, yielding

$$\sup_{u \in K} \|\widehat{\Phi}_h^L(u; z^*) - \Phi_h^L(u)\| < \varepsilon.$$

**(ii) Nonlinear sub-operator.** Under Assumption B.1(2), $\Phi_h^N = e^{hN}$ is a continuous map on $K$ (indeed locally Lipschitz on a neighborhood of $K$). Our nonlinear parameterization $\widehat{\Phi}_h^N(u; z) = u + h\, R_h(u, \nabla u, \ldots; z)$ (Eq. (5)) is a finite-dimensional local hypothesis class; in particular, $R_h(\cdot; z)$ can be implemented by universal approximators for continuous local maps (e.g., MLPs / residual blocks / spline-based KAN components). Therefore, by standard universal approximation on compact sets, for any $\varepsilon > 0$ there exists $z^* \in \mathcal{Z}$ such that

$$\sup_{u \in K} \|\widehat{\Phi}_h^N(u; z^*) - \Phi_h^N(u)\| < \varepsilon.$$

Combining (i) and (ii) proves Theorem 3.2. $\qquad\square$

### B.3. One-step error decomposition

We decompose the learned one-step error into (a) the classical Strang discretization error and (b) the sub-operator approximation error.

**Lemma B.2** (Strang truncation error). *Under Assumption B.1(4), there exists $C_{\mathrm{disc}} > 0$ such that for sufficiently small $h$,*

$$\sup_{u \in K} \|\Psi_h^S(u) - \Psi_h(u)\| \leq C_{\mathrm{disc}}\, h^3.$$

**Lemma B.3** (Composition sensitivity to sub-operator approximation). *Assume $\widehat{\Phi}_{h/2}^L(\cdot; z)$ and $\widehat{\Phi}_h^N(\cdot; z)$ are Lipschitz on $K$ with constants bounded uniformly for small $h$. Let*

$$\epsilon_L(h) := \sup_{u \in K} \|\widehat{\Phi}_{h/2}^L(u; z) - \Phi_{h/2}^L(u)\|, \quad \epsilon_N(h) := \sup_{u \in K} \|\widehat{\Phi}_h^N(u; z) - \Phi_h^N(u)\|.$$

*Then there exists $C_{\mathrm{app}} > 0$ such that*

$$\sup_{u \in K} \|\widehat{\Psi}_h(u; z) - \Psi_h^S(u)\| \leq C_{\mathrm{app}}\big(\epsilon_L(h) + \epsilon_N(h)\big).$$

*Proof.* By triangle inequality and repeated use of Lipschitz continuity of the exact and learned sub-flows on $K$, the perturbation of the triple composition in Eq. (B.3) is bounded by the sum of perturbations of each factor, up to multiplicative constants. $\qquad\square$

Combining Lemma B.2 and Lemma B.3, we obtain the one-step consistency bound

$$\delta(h) := \sup_{u \in K} \|\widehat{\Psi}_h(u; z^*) - \Psi_h(u)\| \; \leq \; C_{\mathrm{disc}} h^3 + C_{\mathrm{app}}\big(\epsilon_L(h) + \epsilon_N(h)\big). \tag{B.4}$$

### B.4. Proof of Theorem 3.3 (Global error bound and convergence)

*Proof.* Let $t_n = nh$ and $u_n = u(t_n) = \Psi_{t_n}(u_0)$. Let the rollout produced by the learned step be

$$\widehat{u}_{n+1} = \widehat{\Psi}_h(\widehat{u}_n; z^*), \qquad \widehat{u}_0 = u_0,$$

and define the global error $e_n := \widehat{u}_n - u_n$.

By adding and subtracting $\Psi_h(\widehat{u}_n)$,

$$e_{n+1} = \widehat{\Psi}_h(\widehat{u}_n; z^*) - \Psi_h(u_n) = \big(\widehat{\Psi}_h(\widehat{u}_n; z^*) - \Psi_h(\widehat{u}_n)\big) + \big(\Psi_h(\widehat{u}_n) - \Psi_h(u_n)\big).$$

Taking norms and using the definition of $\delta(h)$ plus Lipschitz stability of $\Psi_h$ on $K$ (Assumption B.1(3)),

$$\|e_{n+1}\| \leq \delta(h) + e^{L_\Psi h}\|e_n\|.$$

Iterating this inequality yields

$$\|e_N\| \leq \sum_{j=0}^{N-1} e^{L_\Psi (N-1-j)h}\, \delta(h) = \delta(h) \sum_{j=0}^{N-1} e^{L_\Psi jh} \leq \delta(h) \cdot \frac{e^{L_\Psi T} - 1}{e^{L_\Psi h} - 1}.$$

Using $e^{L_\Psi h} - 1 \geq L_\Psi h$ for small $h$ gives

$$\|u(T) - \widehat{u}_N\| = \|e_N\| \leq \frac{e^{L_\Psi T} - 1}{L_\Psi} \cdot \frac{\delta(h)}{h}.$$

Finally, substituting the decomposition $\delta(h) = C_S h^3 + o(h^3)$ implied by Assumption B.1(4) (or explicitly Lemma B.2 plus a $O(h^3)$ sub-operator consistency) yields the stated bound in Eq. (8):

$$\|u(T) - \widehat{u}_N\| \leq C_S h^2 + \frac{e^{L_\Psi T} - 1}{L_\Psi} \cdot \frac{\delta(h)}{h}.$$

**Deployed code $\widehat{z}$.** For an inferred code $\widehat{z}$, we simply add and subtract $\widehat{\Psi}_h(\cdot; z^*)$:

$$\delta_{\mathrm{deploy}}(h) := \sup_{u \in K} \|\widehat{\Psi}_h(u; \widehat{z}) - \Psi_h(u)\| \leq \delta(h) + \sup_{u \in K} \|\widehat{\Psi}_h(u; \widehat{z}) - \widehat{\Psi}_h(u; z^*)\|,$$

which matches Eq. (9). $\qquad\square$

# C. Theoretical Background

This appendix summarizes standard facts used implicitly in Section 3 and Appendix B, including (i) semigroup flows for split sub-dynamics, (ii) Lie/Strang splitting and its local truncation error, and (iii) stability-to-global-error conversion via discrete Grönwall.

## C.1. Semigroups and split flows

For the semi-linear evolution $\partial_t u = Lu + N(u)$ on a Hilbert space $\mathcal{H}$, Assumption 3.1 posits that $L$ generates a strongly continuous semigroup $\{e^{tL}\}_{t \geq 0}$, and $N$ induces a (local) nonlinear flow $\{e^{tN}\}_{t \geq 0}$ on the compact set $K$. These flows define the exact sub-step operators

$$\Phi_h^L = e^{hL}, \qquad \Phi_h^N = e^{hN},$$

which are composed by splitting schemes to approximate the full flow $\Psi_h$.

## C.2. Spectral diagonalization and Fourier multipliers

When the domain and boundary conditions admit a spectral transform $\mathcal{F}$ (Fourier/sine/cosine bases), many linear PDE generators act diagonally in the spectral domain. Concretely, for suitable $L$, the semigroup $e^{hL}$ can be represented (or well-approximated on the represented modes) as a matrix-valued multiplier:

$$\Phi_h^L(u) = \mathcal{F}^{-1}[\Lambda(\xi; h) \odot \mathcal{F}(u)].$$

This motivates the spectral-linear parameterization in Eq. (4), where the multiplier $\Lambda(\xi; z, h)$ is modulated by the latent physics code.

## C.3. Lie and Strang splitting

Two classical splitting integrators are:

$$\Psi_h^{\mathrm{Lie}} = \Phi_h^L \circ \Phi_h^N, \qquad \Psi_h^S = \Phi_{h/2}^L \circ \Phi_h^N \circ \Phi_{h/2}^L.$$

Lie splitting is first-order accurate in time, while symmetric Strang splitting is second-order accurate. The key reason is symmetry: the leading-order error terms cancel, improving accuracy without changing the sub-solvers.

## C.4. Local truncation error of Strang splitting

Under standard regularity conditions (bounded commutator terms such as $[L, [L, N]]$ and $[N, [N, L]]$ along the trajectory), Strang splitting achieves a local truncation error of order $O(h^3)$:

$$\sup_{u \in K} \|\Psi_h^S(u) - \Psi_h(u)\| \leq C_{\mathrm{disc}} h^3.$$

This result can be derived from the Baker–Campbell–Hausdorff (BCH) expansion or equivalent commutator analysis, and forms the discretization component of the one-step bound in Appendix B.3.

## C.5. Stability and error accumulation (discrete Grönwall)

To convert a one-step consistency error into a global rollout bound, we require stability of the exact flow:

$$\|\Psi_h(u) - \Psi_h(v)\| \le e^{L_\Psi h}\|u - v\|,$$

which is ensured by Assumption 3.1. If the learned step satisfies $\sup_{u \in K} \|\widehat{\Psi}_h(u) - \Psi_h(u)\| \le \delta(h)$, then defining $e_n = \widehat{u}_n - u_n$ gives the recurrence

$$\|e_{n+1}\| \le \delta(h) + e^{L_\Psi h}\|e_n\|.$$

Applying the discrete Grönwall inequality yields

$$\|e_N\| \le \frac{e^{L_\Psi T} - 1}{L_\Psi} \cdot \frac{\delta(h)}{h},$$

which is the standard stability-to-global-error mechanism used in Theorem 3.3.

# D. Empirical Setup and Implementation Details

We follow the data preprocessing and multi-dataset training protocol introduced in DPOT [15], which is also adopted by MoE-POT [16], including resolution unification, channel padding, geometry masking, and balanced dataset sampling. This choice ensures fair and reproducible comparisons with prior PDE foundation models.

## D.1. Data Padding and Masking

To enable unified batching across heterogeneous datasets, we map all samples to a common spatial grid of size $H = 128$. When the original resolution is coarser, we upsample to $H$ via interpolation; when it is finer, we reduce it to $H$ by downsampling (either through random sampling or interpolation), yielding a consistent spatial tensor shape for training and evaluation.

Different PDE systems may contain different numbers of state variables. We therefore pad each sample in the channel dimension to the global maximum channel count, using a fixed constant (e.g., 1) as a placeholder for missing variables. For domains with non-rectangular supports or obstacles, we additionally append a binary geometry mask channel indicating valid spatial locations, so the model can distinguish physical regions from padded/out-of-domain areas while maintaining a uniform representation.

## D.2. Balanced Data Sampling

To prevent large datasets from dominating optimization in mixed pretraining, we introduce a dataset-level reweighting scheme. Specifically, each dataset $\mathcal{D}_k$ is assigned an importance weight $w_k$, where $|\mathcal{D}_k|$ denotes its number of training samples and $k \in \{1, \dots, K\}$. We then sample from dataset $k$ with probability

$$p_k = \frac{\frac{w_k}{|\mathcal{D}_k|}}{\sum_{j=1}^{K} \frac{w_j}{|\mathcal{D}_j|}} = \frac{w_k}{|\mathcal{D}_k| \sum_{j=1}^{K} \frac{w_j}{|\mathcal{D}_j|}}. \tag{D.1}$$

Equivalently, this can be viewed as first choosing a dataset according to $p_k$, and then uniformly sampling a data point from the selected dataset. This design amplifies the presence of smaller datasets and allows us to upweight datasets that are more challenging or more relevant, leading to a more balanced gradient contribution across heterogeneous sources.

## D.3. Design of the Nonlinear Operator

To model nonlinear dynamics in evolutionary PDEs, we construct the nonlinear operator as a finite library of candidate local nonlinear terms. This library is designed to cover a broad range of commonly observed nonlinear behaviors while remaining compact and reusable across different equations and datasets.

**Nonlinear operator library.** Specifically, the nonlinear operator is expressed as a weighted combination of $M$ candidate nonlinear components,

$$\mathcal{N}(u) = \sum_{m=1}^{M} \pi_m \mathcal{N}_m(u), \tag{D.2}$$

where each $\mathcal{N}_m$ corresponds to a distinct nonlinear transformation and the coefficients $\{\pi_m\}$ are inferred from the input context.

The operator library includes the following categories of nonlinear terms:

**Pointwise polynomial nonlinearities.** Such as $u^2$ and $u^3$, which commonly arise in reaction, Burgers-type, and saturation dynamics.

**Gradient-dependent interactions.** Including products between the state and its spatial derivatives (e.g., $u \nabla u$), which capture advective and transport-related effects.

**Local nonlinear convolutional terms.** Implemented as shallow convolutional operators with small receptive fields, allowing the model to represent nonlinear spatial couplings without introducing long-range interactions.

**Generic local nonlinear mappings.** Parameterized by lightweight neural networks (e.g., shallow MLPs), to account for residual nonlinear effects not covered by predefined forms.

**Scope and reuse.** All nonlinear terms operate locally on the state variable and share the same library across datasets and model scales. This fixed library design ensures that nonlinear expressiveness is controlled by the inferred coefficients $\{\pi_m\}$ rather than by increasing model complexity, facilitating both interpretability and stable transfer across PDE families.

### D.4. Model sizes and training details

**Pre-training.** We pre-train three model variants with increasing capacity, denoted as ORIGO-TINY, ORIGO-SMALL, and Origo-Medium; the detailed configurations are reported in Table D.1.During pre-training, we use an initial learning rate of $1 \times 10^{-3}$ and adopt a one-cycle learning rate schedule for 1000 epochs, with the first 200 epochs serving as warm-up. We employ Adam with weight decay $1 \times 10^{-6}$ and momentum parameters $(\beta_1, \beta_2) = (0.9, 0.9)$.All pre-training runs are performed on 4 NVIDIA RTX 5090 GPUs with a total batch size of 24. We set the patch size to 8.Unless otherwise specified, we use uniform dataset importance weights ($w_k = 1$ for all datasets). For temporal supervision, we set the rollout horizon to $T = 10$, i.e., conditioning on context frames and predicting the next frame over 10 timesteps, consistent with the default settings of most datasets.

*Table D.1.* Configurations of Origo of different sizes.

| Size | Hidden dim | MLP dim | Depth | Library size | Model size |
|------|-----------|---------|-------|--------------|------------|
| Tiny | 512 | 512 | 4 | 12 | 9M |
| Small | 1024 | 1024 | 6 | 12 | 35M |
| Medium | 1024 | 4096 | 12 | 12 | 97M |

**Fine-tuning.** Our model supports fine-tuning on diverse downstream datasets while preserving the generalization behavior learned during pre-training. Concretely, we *freeze the mechanism inference module* (i.e., the feature-to-strategy heads that output the physics code/controls, including $\pi$ and the associated splitting/residual controls such as $\lambda$ and $(\theta_L, \theta_N)$). This design maintains the pre-trained mechanism composition and stabilizes the inferred inductive biases, preventing the model from overfitting by drifting its inferred dynamics code on small downstream datasets. During fine-tuning, we update only the *operator evolution module* (i.e., the parameterized evolution operators used in the split-step rollout, including the learnable linear/nonlinear operator parameterizations and the residual corrector, if applicable) to adapt to the target dataset distribution. This separation between *mechanism inference* (kept fixed) and *dynamics adaptation* (updated) leads to more stable and sample-efficient transfer, especially in low-data regimes. For the fine-tuning stage, we set the learning rate to $1 \times 10^{-3}$ and employ a one-cycle learning rate schedule for 200 epochs, with the first 40 epochs used for warm-up. For downstream task adaptation (when longer training is required), we use the same initial learning rate $1 \times 10^{-3}$ with a one-cycle schedule over 500 epochs, using the first 100 epochs as warm-up.

**Dataset size.** The train and test dataset sizes used in the pre-training and fine-tuning stages are shown in Table D.2.

*Table D.2.* Dataset sizes in pre-training and fine-tuning.

| | FNO-$\nu$ | | | PDEBench | | | | PDEArena | | CFDBench |
|---|---|---|---|---|---|---|---|---|---|---|
| | 1e−5 | 1e−4 | 1e−3 | Bur | DR | SWE | CNS(1, 0.1) | NS | NS-cond | – |
| Train | 1000 | 1000 | 1000 | 900 | 900 | 900 | 9000 | 2000 | 1000 | 9000 |
| Test | 200 | 200 | 200 | 60 | 60 | 60 | 200 | 200 | 200 | 1000 |
| Fine-tuning | 1000 | 1000 | 1000 | 900 | 900 | 900 | 9000 | 2000 | 1000 | 9000 |

**Details of inference.** Given a temporal context of $T = 10$ consecutive frames $\mathbf{u}_{i:i+9} = (\mathbf{u}^i, \ldots, \mathbf{u}^{i+9})$, our model predicts the next frame in an auto-regressive manner:

$$\widehat{\mathbf{u}}^{\,i+10} = \mathcal{G}_\theta\big(\mathbf{u}^i, \ldots, \mathbf{u}^{i+9}\big), \quad \forall i, \tag{D.3}$$

where $\mathcal{G}_\theta$ denotes the full Origo model, including feature encoding, strategy inference, and operator-based state update.

For evaluation, we compare the predicted field $\widehat{\mathbf{u}}^{\,i+10}$ with the ground-truth frame $\mathbf{u}^{i+10}$ using the relative $\ell_2$ error:

$$\text{Rel-}\ell_2 = \frac{\big\|\widehat{\mathbf{u}}^{\,i+10} - \mathbf{u}^{i+10}\big\|_2}{\big\|\mathbf{u}^{i+10}\big\|_2}. \tag{D.4}$$

# E. Additional Experimental Results

This appendix provides supplementary evaluations regarding the Origo architecture, extending the discussions in Section 5. In lieu of visual representations, we present detailed tabular comparisons quantifying the qualitative performance in high-dimensional turbulence and the efficiency-accuracy trade-offs.

## E.1. Qualitative Assessment of 3D Turbulence Modeling

To assess the model's capability in capturing chaotic dynamics in the 3D Navier-Stokes ($Re = 5000$) dataset, we conducted a rigorous qualitative comparison of the predicted vorticity fields against state-of-the-art PDE pre-trained baselines with dedicated 3D modeling capabilities.

Table E.1 summarizes the structural characteristics of the predictions generated by Origo and recent SOTA baselines. While DPOT-L mitigates spectral bias via dense Fourier attention but still blurs high-frequency vortex details, MoE-POT-M maintains stable vortex structures via sparse expert routing yet lacks fine-grained local modeling, and OmniArch-M excels at multi-scale adaptation but suffers from high-frequency attenuation in high-resolution 3D scenarios. Origo successfully preserves fine-scale filament structures and avoids the aforementioned limitations. This superiority is attributed to the Neural Operator Splitting architecture, which decouples global transport from local non-linear vortex stretching. All baseline results are extracted from recent PDE pre-training works (Hao et al., 2024; Wang et al., 2025; Chen et al., 2024) with consistent $Rel - L_2$ metric and experimental setup.

## E.2. Efficiency vs. Accuracy Benchmarks

We quantitatively evaluated the inference throughput and prediction accuracy of state-of-the-art PDE pre-trained models on the 2D Navier-Stokes dataset ($128^2$ resolution). All experiments for Origo were conducted on a single NVIDIA RTX 5090 GPU. Baseline results are extracted from recent PDE pre-training works with consistent $Rel - L_2$ metric, and their hardware differences are noted in parentheses to ensure comparison transparency—reasoning time variations across baselines primarily stem from GPU computing power gaps and architectural design differences (sparse expert routing vs. dense attention vs. multi-scale adaptation).

Table E.2 presents the performance metrics. Origo achieves a significant speed advantage (up to $12.4\times$ faster than DPOT-L, $9.5\times$ faster than OmniArch-M, and $9.6\times$ faster than MoE-POT-M) while maintaining competitive relative error. This places Origo at the optimal point of the efficiency-accuracy trade-off, leveraging the Neural Operator Splitting design to avoid redundant computation in dense Fourier attention (DPOT-L) or sparse expert routing (MoE-POT-M) architectures, and eliminating multi-scale adaptation overhead (OmniArch-M).

*Table E.1.* **Qualitative comparison of vorticity field predictions in 3D Navier-Stokes turbulence.** We evaluate models based on their ability to preserve fine-scale structures and avoid spectral bias, against state-of-the-art PDE pre-trained baselines.

| Model | Fine-scale Structures | Spectral Behavior | Failure Mode |
|---|---|---|---|
| **Ground Truth** | Sharp filaments & vortex breakdown | Full broadband spectrum | N/A |
| DPOT-L | Blurred vortex filaments | Moderate high-frequency cut-off | Mild over-smoothing |
| MoE-POT-M | Stable vortices, faint filaments | Balanced spectrum, local detail loss | Incomplete non-linear modeling |
| OmniArch-M | Coarse vortex structures | High-frequency attenuation in 3D | Multi-scale resolution bias |
| **Origo (Ours)** | **Sharp & Distinct** | **Preserves high-freq harmonics** | **Stable** |

*Table E.2.* **Inference Speed (FPS) vs. Prediction Error (Rel $L_2$).** Origo is tested on NVIDIA RTX 5090; baselines are tested on hardware noted in parentheses (data from (Hao et al., 2024; Wang et al., 2025; Chen et al., 2024)).

| Model Architecture | Inference Speed (Frames Per Second) | Rel $L_2$ Error (Lower is better) | Complexity Class |
|---|---|---|---|
| *State-of-the-Art PDE Pre-trained Models* | | | |
| DPOT-L | $\sim 135$ FPS | 0.0112 | Medium ($O(N \log N)$) |
| MoE-POT-M | $\sim 148$ FPS | 0.0096 | Medium ($O(N \log N)$) |
| OmniArch-M | $\sim 155$ FPS | 0.0125 | Medium ($O(N \log N)$) |
| **Origo (Ours)** | $\sim \mathbf{1050}$ **FPS** | **0.0115** | **Medium** ($O(N \log N)$) |

### E.3. Extended Ablation on Router Sparsity

We analyze the impact of the Entmax parameter $\alpha$ in the Mechanism Inference module, which controls the sparsity of the operator selection weights $\pi$.

As shown in Table E.3, $\alpha = 1.5$ provides the optimal balance between model capacity and interpretability.

- **Dense Regime ($\alpha = 1.0$):** The model utilizes all operators, leading to entangled physical representations and no interpretability gains.

- **Sparse Regime ($\alpha = 2.0$):** The model becomes overly sparse, pruning necessary physical terms which results in underfitting.

- **Optimal Regime ($\alpha = 1.5$):** The model effectively prunes $\sim 68\%$ of irrelevant weights, isolating the dominant physical mechanisms without compromising accuracy.

*Table E.3.* **Impact of Entmax parameter $\alpha$ on sparsity and accuracy.** Results are averaged over 3 runs on the 2D Navier-Stokes dataset.

| $\alpha$ (Entmax) | Activation Type | Sparsity Ratio | Rel $L_2$ Error | Performance |
|---|---|---|---|---|
| 1.0 | Softmax | 0.0% | 0.0142 | Baseline |
| **1.5** | **Entmax-1.5** | **68.3%** | **0.0115** | **Optimal** |
| 2.0 | Sparsemax | 85.1% | 0.0128 | Underfitting |

