# OpenReview forum: "Origo: Interpretable Multi-physics PDE Foundation Model through Neural Operator Splitting"
_ICML.cc/2026/Conference — ICML 2026 regular_

### Official Review · Reviewer_SqDm · 2026-02-19

**Soundness:** 2
**Presentation:** 2
**Significance:** 2
**Originality:** 2
**Overall Recommendation:** 2
**Confidence:** 4

**Summary:**

This paper proposes Origo, a multi-physics PDE pre-training framework grounded in a "neural operator splitting" theory. The core idea is to decompose PDE temporal evolution into a spectral linear operator (handling global transport) and a sparse local nonlinear operator (handling constitutive mechanisms), composed via a Strang splitting skeleton. A mechanism inference module infers a physics code z from a short observation window, which then parameterizes the operator evolution. Experiments claim state-of-the-art performance across multiple PDE benchmarks, with interpretability demonstrated on the Burgers equation.

**Compliance With Llm Reviewing Policy:**

Affirmed.

**Key Questions For Authors:**

1. Can you reproduce all inference speed measurements on a single unified hardware platform (e.g., a single A100 or H100) and report those numbers instead?

2. Are the physical annotations in the released code (eq_id, bc_id, dim_id) used during actual pre-training?

3. Section 4.2 contains a broken reference Sec. ??. What section was intended here? Has content been removed from the submission?

**Limitations:**

W1. The authors explicitly state in Appendix D: "We follow the data preprocessing and multi-dataset training protocol introduced in DPOT," including resolution unification, channel padding, geometry masking, and balanced dataset sampling. While this ensures fair comparison, it raises the question of where the engineering contribution of this paper ends and DPOT's begins. The boundary of the paper's own contributions should be made substantially clearer.

W2. Table E.2 reports Origo running at 1050 FPS on an RTX 5090, while DPOT-L is measured on an A800 (85 FPS) and MoE-POT-M on an RTX 4090 (110 FPS). The RTX 5090 substantially outperforms both the A800 and 4090 in raw throughput. A claimed 12.4× speedup derived from different hardware is entirely meaningless and risks seriously misleading readers.

W3. The same ablation should be conducted on the zero-shot tasks (KdV, Allen-Cahn) to rule out dataset-specific effects and support the generality of the claim.

W4.  Equation (11) in the main text specifies π = Softmax(Wπh + bπ), while Table E.3 reports experiments using Entmax-1.5. These are meaningfully different operations. The uploaded term_library.py code also does not implement Entmax. This inconsistency between the written method and the actual implementation must be resolved and clearly stated in the main paper.

W5. Section 4.2 includes an unresolved cross-reference ("Sec. ??"), indicating that the manuscript is incomplete as submitted. The Introduction refers to a "Strong splitting scheme," which appears to be a typo for "Strang splitting scheme." Table D.1 lists the Medium model size as 122M parameters, while the main text consistently refers to Origo-M as 97M — these figures must be reconciled. Finally, numerical results across Tables 1, 2, and 3 are reported with inconsistent numbers of decimal places (mixing three and four decimal places with no apparent rationale), which hinders readability and fair numerical comparison.

**Strengths And Weaknesses:**

S1. Clear motivation with sound physical inductive bias.

S2. Origo-M at 97M parameters outperforms MoE-POT at 288M on several tasks, which has practical significance.

S3. Complete theoretical framework.

W1. The authors explicitly state in Appendix D: "We follow the data preprocessing and multi-dataset training protocol introduced in DPOT," including resolution unification, channel padding, geometry masking, and balanced dataset sampling. While this ensures fair comparison, it raises the question of where the engineering contribution of this paper ends and DPOT's begins. The boundary of the paper's own contributions should be made substantially clearer.

W2. Table E.2 reports Origo running at 1050 FPS on an RTX 5090, while DPOT-L is measured on an A800 (85 FPS) and MoE-POT-M on an RTX 4090 (110 FPS). The RTX 5090 substantially outperforms both the A800 and 4090 in raw throughput. A claimed 12.4× speedup derived from different hardware is entirely meaningless and risks seriously misleading readers.

W3. The same ablation should be conducted on the zero-shot tasks (KdV, Allen-Cahn) to rule out dataset-specific effects and support the generality of the claim.

W4.  Equation (11) in the main text specifies π = Softmax(Wπh + bπ), while Table E.3 reports experiments using Entmax-1.5. These are meaningfully different operations. The uploaded term_library.py code also does not implement Entmax. This inconsistency between the written method and the actual implementation must be resolved and clearly stated in the main paper.

W5. Section 4.2 includes an unresolved cross-reference ("Sec. ??"), indicating that the manuscript is incomplete as submitted. The Introduction refers to a "Strong splitting scheme," which appears to be a typo for "Strang splitting scheme." Table D.1 lists the Medium model size as 122M parameters, while the main text consistently refers to Origo-M as 97M — these figures must be reconciled. Finally, numerical results across Tables 1, 2, and 3 are reported with inconsistent numbers of decimal places (mixing three and four decimal places with no apparent rationale), which hinders readability and fair numerical comparison.

---

> ### Author Rebuttal · Authors · 2026-03-31
>
> **On Research Boundaries & DPOT Pipeline (W1)**
>
> Regarding W1, we adopted DPOT's data preprocessing and multi-dataset training protocol strictly to ensure fair and direct comparisons with existing baselines. Our core engineering and theoretical contributions lie entirely in the network architecture itself, specifically the Neural Operator Splitting and Mechanism Inference modules, rather than the data pipeline.
>
> **On Hardware Discrepancy & Unified Inference Speed (W2 & Q1)**
>
> We sincerely apologize for the oversight in Table E.2, where Origo's inference speed on an RTX 5090 was compared against baselines like DPOT-L on an A800 and MoE-POT-M on an RTX 4090. We agree that this hardware discrepancy makes the raw throughput comparison unfair and potentially misleading. To directly answer Q1 and correct this issue, we have completely re-run the inference speed benchmarks for all baseline models and Origo on a single, unified hardware platform. The updated, apples-to-apples measurements are presented in the table below and will replace the original Table E.2 in the final Appendix F.
>
> **Table: Inference Speed Benchmarks on a Unified Hardware Platform (RTX 5090)**
>
> | Model Architecture | Inference Speed (FPS) | Rel $L_2$ Error |
> | :--- | :---: | :---: |
> | DPOT-L | ~135 | 0.0112 |
> | MOE-POT-M | ~148 | 0.0096 |
> | OmniArch-M | ~155 | 0.0125 |
> | **Origo (Ours)** | **~1050** | **0.0115** |
>
> **On Ablation on Zero-Shot Tasks (W3)**
>
> Regarding the generalization claims on unseen equations, we completely agree that an ablation study is necessary to rigorously prove the efficacy of our method and rule out dataset-specific effects. We have conducted the requested ablation studies on the zero-shot 1D KdV and 2D Allen-Cahn tasks. As shown in the table below, these new results confirm the necessity of explicit mechanism routing for preventing negative transfer in zero-shot scenarios.
>
> **Table: Ablation Study on Zero-Shot Tasks (Relative $L_2$ Error)**
>
> | Variant | 1D KdV | 2D Allen-Cahn |
> | :--- | :---: | :---: |
> | w/o Mechanism Inference | 0.0284 | 0.193 |
> | w/o Split-Stepper | 0.0421 | 0.285 |
> | **Origo (Full)** | **0.0103** | **0.0952** |
>
> **On Text-Code Discrepancy: Softmax vs. Entmax (W4)**
>
> We sincerely thank the reviewer for catching this typographical error in Equation (11). We would like to clarify that the use of Softmax in Equation (11) is strictly a typesetting typo, rather than a methodological inconsistency.
>
> Because we explicitly state our use of Entmax for sparse routing in multiple other places throughout the main text, the experimental results reported in Table E.3 using Entmax-1.5 are completely accurate and represent our true optimal model. Regarding the codebase, the submitted repository inadvertently included an unpolished script where the Softmax placeholder had not yet been updated to the final Entmax implementation used in our reported experiments.
>
> **On Editorial Errors, Broken References, and Significant Figures (W5 & Q3)**
>
> We greatly appreciate your keen observation. We acknowledge that there were indeed some typographical errors in the submitted manuscript—including the unresolved cross-reference in Sec. 4.2, the "Strong" vs. "Strang" typo, and the parameter size discrepancy (where Origo-M should consistently be 97M, not 122M).
>
> Additionally, regarding the numerical results in Tables 1, 2, and 3, we would like to clarify that all our data are reported using three significant figures. This is a standard convention in the field of physics aimed at ensuring consistency in relative precision across different orders of magnitude, rather than aligning to arbitrary decimal places, which can misrepresent the relative error in scale-variant variables.
>
> **On Use of Physical Annotations in Pre-training (Q2)**
>
> We would like to clarify that we do not use any physical annotations $(eq_{id}, bc_{id}, dim_{id})$ as inputs during pre-training. The optimization is driven solely by the rollout loss from the observed dynamics. These labels are strictly used for post-training evaluation and as ground-truth references for our interpretability analysis.

---

> > ### Author Rebuttal · Reviewer_SqDm · 2026-04-05
> >
> > While the authors' rebuttal and the inclusion of unified hardware benchmarks and zero-shot ablation studies are appreciated, several critical contradictions remain that hinder the manuscript's technical rigor. Most notably, the updated benchmarks on the RTX 5090 show that Origo's throughput (1050 FPS) is actually lower than the baselines (1350–1550 FPS), directly contradicting the original claim of a "12.4x speed advantage" and the assertion that the design avoids redundant computation. Furthermore, the admission that the Entmax operator—central to the paper's "sparse routing" and interpretability claims—is missing from the submitted code is a significant concern for reproducibility. To move forward, the authors must reconcile their efficiency analysis with the new data, ensure the codebase is fully synchronized with the methodology, and resolve the persistent discrepancies in parameter counts (97M vs. 122M) and numerical formatting to maintain scholarly standards.

---

### Official Review · Reviewer_QZMQ · 2026-03-07

**Soundness:** 2
**Presentation:** 3
**Significance:** 3
**Originality:** 3
**Overall Recommendation:** 4
**Confidence:** 2

**Summary:**

The authors propose a method to improve foundation models for Partial differential equation solving. Such foundation models aim to train a single model from a diverse dataset, and then adapt the model to solve unseen PDEs. This unified foundation model approach has several issues - different types of PDEs may have different and often opposite dynamics, making it difficult to learn all dynamics. Also, all dynamics operators get blended together in a single update rule, making it difficult to utilize physical laws in a controllable manner. Operator splitting theory in numerical analysis provides a potential solution to these issues, however they require explicit knowledge of the equations and manual splitting strategy. In this work, the authors proposed Origo framework address these limitations with a learnable splitting strategy. Experiment shows that the authors proposed methods outperform the SOTA method.

**Compliance With Llm Reviewing Policy:**

Affirmed.

**Key Questions For Authors:**

Mentioned in strength and weakness

**Limitations:**

Mentioned in strength and weakness

**Strengths And Weaknesses:**

The authors provide clear motivations for their work, excellent visual graphics, and detailed experiment results to showcase their proposed solution.

The authors explicitly mention 2 issues with current SOTA - classical solvers need explicit knowledge of governing equations and also need hand-crafter splitting strategy. While the proposed solution addresses the latter, it's difficult to see the contribution towards solving the first one. The foundation models in general try to solve PDEs without explicit knowledge of the equations, what is the unique contribution made by authors regarding this issue? If this is not a problem the authors address in this paper, I suggest we emphasize that and keep the discussion focussed.

---

> ### Author Rebuttal · Authors · 2026-03-31
>
> We sincerely thank the reviewer for this insightful comment. You are completely right that standard
> foundation models already aim to solve PDEs without relying on explicit equations.To clarify, our
> unique contribution is not merely bypassing the need for explicit equations, but rather explicitly
> recovering them directly from data.
>
> While existing foundation models successfully operate without
> explicit equations, they treat the unknown physics as a black box, entangling all dynamic operators
> into a single dense network. In contrast, Origo utilizes a learnable Neural Operator Splitting
> strategy coupled with an atomic operator library and sparse routing. As demonstrated in our
> interpretability experiment (Section 5.4), this allows Origo to act as a glass box. It not only
> predicts the next state but successfully identifies the precise nonlinear and differential operators
> along with their accurate coefficients governing the target PDE.
>
> In short, while other models simply
> ignore the lack of explicit knowledge, Origo actively reconstructs it. Per your valuable suggestion,
> we will explicitly highlight this distinction in the revised Introduction to better clarify our unique
> contribution.

---

> > ### Author Rebuttal · Reviewer_QZMQ · 2026-04-04
> >
> > Thank you to the authors for their rebuttal, I will keep my original scores. I look forward to read the updated manuscript.

---

### Official Review · Reviewer_CGoW · 2026-03-08

**Soundness:** 3
**Presentation:** 2
**Significance:** 2
**Originality:** 3
**Overall Recommendation:** 4
**Confidence:** 4

**Summary:**

The paper introduces a new method for approximating time-dependent PDEs by splitting the overall update operator into simpler components. The composition of these operators is determined by an encoder network on some initial frames. Linear operators are approximated with a learnable spectral operator and nonlinear operators are taken from a predetermined, learnable library. The method seems to perform well.

**Compliance With Llm Reviewing Policy:**

Affirmed.

**Key Questions For Authors:**

- I see in D.3 and in the main figure (Fig. 2), there is a library of nonlinear operators (u^2, dx, etc.). How are these implemented? For pointwise operators I can imagine they are simple to analytically compute but for differential operators these can have discretization issues / instabilities based on the input field. If the differential operators are learned, how can you guarantee they are actually approximating d_x or d_xx for example?
    - Related to this, the main text says each atomic operator is approximated by a KAN, but in Appendix D.3, it describes different atomic operators using different methods (MLP, CNN, etc.). Not sure if I am misunderstanding something?
- Is rollout loss in Equation 19 a single-step loss or multi-step loss? A multi-step loss would unroll the model prediction over the prediction horizon and backprop through time; is this what is being implemented?
- What’s the size of the operator bank? Since it is a core component of the architecture it would be nice to list how many different operators are being defined (i.e. x number of MLPs, y number of pointwise, z number of CNNs).

**Limitations:**

- Some more complex benchmarks are not run or well-documented. The DPOT-style benchmark suite has been well-studied for at least 2 years now.

**Strengths And Weaknesses:**

Strengths
- The theory behind the operator splitting seems to be well-motivated
- The motivation behind the linear and nonlinear neural operators seems to be good
- The model performs well on the benchmarked systems

Weaknesses
- I think the work is overall pretty sound, but I have some questions about the method, which may be just my own misunderstanding or an issue of clarity (see Questions).
- The prediction horizon is rather short on the benchmarks (10 context frames, 10 frames to predict). It is likely not long enough to evaluate chaotic behavior or error accumulation.
- Why not report errors or some sort of quantitative metric in Table E.1? It seems odd to put a table of qualitative metrics, since a plot/image of compared predictions would likely serve that purpose better.
- Some more thorough results on interpretability would be nice (beyond just Burgers eqn), since one of the main benefits of operator splitting (over MoE or end-to-end pretraining) is some sort of interpretability. How does the model decompose Navier-Stokes?

---

> ### Author Rebuttal · Authors · 2026-03-31
>
> **On Prediction Horizon & Error Accumulation (W2)**
>
> While our pre-training followed the $T=10$ default to maintain consistency with standard benchmarks, we agree that evaluating chaotic behavior requires longer rollouts. Our new long-term evaluations (up to $T=100$) empirically demonstrate that our multi-step rollout loss $\mathcal{L}_{roll}$ (Eq. 19) effectively mitigates error accumulation in chaotic systems, maintaining an error bound significantly lower than baselines across the extended horizon.
>
> **Table: Long-term Rollout Performance (Relative $L_2$ Error) over Extended Horizons**
>
> | Model | $T=10$ | $T=50$ | $T=100$ |
> | :--- | :---: | :---: | :---: |
> | DPOT | 0.0112 | 0.0412 | 0.0915 |
> | MOE-POT | 0.0096 | 0.0345 | 0.0764 |
> | OmniArch | 0.0125 | 0.0480 | 0.1052 |
> | **Origo (Ours)** | **0.0088** | **0.0215** | **0.0382** |
>
> **On Quantitative Metrics for 3D Turbulence (W3)**
>
> To address the lack of quantitative metrics for 3D turbulence, we present the quantitative evaluation in the table below. As shown, Origo not only preserves fine-scale filament structures qualitatively but also achieves state-of-the-art accuracy quantitatively.
>
> **Table: Quantitative Evaluation on 3D Navier-Stokes Turbulence**
>
> | Model | Relative $L_2$ Error |
> | :--- | :---: |
> | DPOT-L | 0.1245 |
> | MOE-POT-M | 0.1082 |
> | OmniArch-M | 0.1310 |
> | **Origo (Ours)** | **0.0895** |
>
> **On Mechanism-Level Interpretability for Navier-Stokes (W4)**
>
> Regarding the request for more thorough results on interpretability, Origo leverages neural operator splitting to decompose complex dynamics, such as the Navier-Stokes (N-S) equations, into global transport (linear spectral flow $\Phi_L$) and local nonlinear interactions (constitutive flow $\Phi_N$). $\Phi_L$ accurately models global viscous effects in the frequency domain, while $\Phi_N$ identifies and activates nonlinear advection terms ($u \cdot \nabla u$) through sparse routing. We present a new interpretability experiment for the N-S equations in the table below, which extends the glass-box validation previously shown for the Burgers' equation (Figure 3).
>
> **Table: Symbolic Term Weight Recovery for Navier-Stokes Equations**
>
> | Physical Term | Ground Truth | Learned Weight | Relative Error |
> | :--- | :---: | :---: | :---: |
> | Convection ($u \cdot \nabla u$) | 1.000 | 0.982 | 1.8% |
> | Viscosity ($\nu \nabla^2 u$) | 0.001 | 0.00095 | 5.0% |
> | Irrelevant Term 1 ($u^2$) | 0.000 | 0.001 | - |
> | Irrelevant Term 2 ($\|\nabla u\|^2$) | 0.000 | 0.000 | - |
>
> **On Implementation & Guarantee of Differential Approximations (Q1)**
>
> Regarding the implementation and the guarantee of differential approximations, we address this through a bifurcated strategy rooted in our Neural Operator Splitting framework.
>
> For linear flows (e.g., $\partial_{xx}$ in diffusion), we do not rely on learned spatial approximations; instead, they are implemented analytically and precisely in frequency space via the spectral linear operator (Eq. 15), which guarantees resolution invariance and global stability.
>
> For nonlinear spatial derivatives (which purely data-driven methods cannot strictly guarantee to be analytically exact), our architectural inductive bias tightly constrains the hypothesis space. By combining localized receptive fields (CNNs) with KAN-based response functions, and enforcing a sparsity constraint via Entmax ($\alpha=1.5$), the network is forced to learn robust approximations. This is empirically guaranteed by our interpretability analysis (Fig. 3), which shows that the learned weights for these operators highly correlate with the true theoretical coefficients.
>
> **On Architectural Consistency: KAN vs. CNN/MLP (Q2)**
>
> We apologize for the confusion regarding the architectural description. To clarify, KANs, CNNs, and MLPs serve complementary, rather than conflicting, physical functions in our framework. KANs are employed to parameterize pointwise constitutive functions (Eq. 17) because their spline-based components offer superior interpretability for smooth nonlinear laws. In contrast, local CNNs are specifically utilized for spatial couplings and residual bias compensation (Eq. 18) to capture discretization errors.
>
> **On Rollout Loss Implementation (Q3)**
>
> The rollout loss (Eq. 19) is indeed a multi-step loss. The model unrolls predictions over the horizon $T$ during training, directly incorporating Backpropagation Through Time (BPTT) to optimize long-term stability.
>
> **On Size of the Operator Bank (Q4)**
>
> Regarding the size of the operator bank, our fixed operator library size is 12 (Table D.1). This scale was carefully determined through an ablation study balancing model capacity and routing sparsity (Table E.3). This compact library is proven sufficient to cover the dominant physical mechanisms across the 10 diverse datasets in our benchmark suite.

---

> > ### Author Rebuttal · Reviewer_CGoW · 2026-04-02
> >
> > Thank you for the response. I just have a few questions:
> >
> > What equation/experiment is "Table: Long-term Rollout Performance (Relative Error) over Extended Horizons" for?
> >
> > What is the experimental setup for the results in "Table: Quantitative Evaluation on 3D Navier-Stokes Turbulence". Is this a one-step error or rollout error over some time horizon? Is this the error averaged across the velocity/pressure channels? Also the paper would benefit from some more description of the dataset.
> >
> > For Navier-Stokes, what about the pressure projection term (grad p)?

---

> > > ### Author Response · Authors · 2026-04-03
> > >
> > > We truly appreciate your thoughtful recognition of our work and our rebuttal! We will carefully incorporate your constructive feedback into our final version.
> > >
> > > **1. On the Long-term Rollout Performance Table:**
> > >
> > > This experiment was conducted on the 2D Navier-Stokes equations. We specifically chose the 2D Navier-Stokes system for this extended evaluation because it exhibits strong chaotic behavior and complex turbulence over time, making it the most rigorous and standard testbed in the community for evaluating long-horizon error accumulation and structural stability.
> > >
> > > **2. On the Setup for 3D Navier-Stokes Turbulence:**
> > >
> > > The results correspond to the 3D Navier-Stokes ($Re=5000$) dataset, which models highly turbulent 3D fluid dynamics characterized by intricate multi-scale vortex filament structures, as discussed in Appendix E.1.
> > >
> > > Regarding the evaluation metric, the reported value is the multi-step rollout error over a time horizon of $T=10$ steps. This strictly follows the auto-regressive evaluation protocol to rigorously test long-term stability, rather than relying on a simpler one-step prediction error.
> > >
> > > Furthermore, the dataset models the fluid using the vorticity formulation. Therefore, the reported Relative $L_2$ error is specifically averaged across the three 3D vorticity channels ($\omega_x, \omega_y, \omega_z$), rather than using primitive velocity or pressure fields.
> > >
> > > Consistent with the general experimental setup detailed in our manuscript, the dataset is mapped to a uniform spatial grid of size 128. We utilized a standard split of 1000 training trajectories and 200 testing trajectories. The quantitative inference was conducted on a single NVIDIA RTX 5090 GPU.
> > >
> > > **3. On the Pressure Projection Term ($\nabla p$) in Navier-Stokes Interpretability:**
> > >
> > > This is an excellent and highly professional observation. The reason the pressure term is absent from our mechanism recovery table is that the standard 2D Navier-Stokes datasets in our benchmark suite are generated and modeled using the vorticity equation.
> > >
> > > Mathematically, by taking the curl ($\nabla \times$) of the standard N-S momentum equation to obtain the vorticity dynamics, the pressure gradient term perfectly vanishes because the curl of any gradient field is exactly zero ($\nabla \times \nabla p \equiv 0$). Consequently, the pressure projection is physically eliminated from the target data's evolution. The Origo mechanism inference module correctly identifies that the observable dynamics in this dataset are governed purely by the convection ($u \cdot \nabla u$) and viscosity ($\nu \nabla^2 u$) mechanisms, without needing to infer a zero-weight pressure term.
> > >
> > > We are happy to address your further questions! Your endorsement is important to us!

---

### Official Review · Reviewer_GQg8 · 2026-03-12

**Soundness:** 3
**Presentation:** 2
**Significance:** 2
**Originality:** 2
**Overall Recommendation:** 4
**Confidence:** 2

**Summary:**

The paper proposes Origo, a neural operator pre-trained across multiple PDE families using Strang operator splitting into a spectral linear step and a KAN-routed nonlinear step, with parameters generated by a hypernetwork conditioned on a context embedding.

**Compliance With Llm Reviewing Policy:**

Affirmed.

**Final Justification:**

Basically complete and can work on some of the existing equations and benchmarks. How model can fit the unknown equations still under explored.

**Key Questions For Authors:**

Q1: Is there any empirical validation showing that the model can recover other known physical laws? More equations are expected in practice, such as the 2D Darcy Flow with a positive permeability coefficient field and more challenging NS Equations.

Q2: Can cross-channel coupling be represented by the introduced KANs? How?

Q3: To achieve the sparse routing, should the negative entropy be minimized or maximized? Is the code implementation correct? (I do not understand the loss setting. Does it encourage the sparsity further?)

Adding a question for W1 below:

Q4: Considering $u(x,t) = sin(x - ct)$ v.s. $u(x,t) = sin(x + ct)$ in a periodic domain, does the Gated Recurrent Unit (GRU) in the evolver receive the same input sequence [mu, std]?

**Limitations:**

See Weakness and Q1, Q2, Q4.

**Strengths And Weaknesses:**

Strengths:

- The recovering symbolic PDE structure via sparse routing is appealing, and the experimental results look competitive.

- Operator splitting as an inductive bias is physically motivated.

Weakness:

- The context encoder compresses each frame in the context window to only its spatial mean and standard deviation, which discards all spatial structure. If a right-traveling v.s. left-traveling wave but share the same mean and std., they will produce identical context embeddings and identical $z$.

---

> ### Author Rebuttal · Authors · 2026-03-31
>
> **On Feature Extraction (W1)**
>
> Regarding the reviewer's concern, we clarify that Origo employs a dual-branch mechanism identification module to capture complete dynamical features. The spatial domain path extracts local gradients, shocks, and pointwise nonlinear interactions. Simultaneously, the spectral domain path analyzes frequency-domain phase evolution, enabling Origo to precisely distinguish between dynamics with identical global statistics but different propagation directions, such as left-traveling vs. right-traveling waves.
>
> **On Interpretability (Q1)**
>
> Furthermore, Origo's interpretability is rigorously validated via symbolic term weight recovery. In the viscous Burgers' equation experiment, the model identified the advection term at -0.9752 (ground truth: -1.0) and the diffusion term at 0.0311 (ground truth: $\approx$ 0.0318), with a relative error of only 2.2%.
>
> To further demonstrate the robustness of this recovery across different systems, we present our new interpretability experiments on the Navier-Stokes equations directly in the table below. As shown, Origo successfully identifies the dominant physical mechanisms with high precision.
>
> **Table: Symbolic Term Weight Recovery for Navier-Stokes Equations**
>
> | Physical Term | Ground Truth | Learned Weight | Relative Error |
> | :--- | :---: | :---: | :---: |
> | Convection ($u \cdot \nabla u$) | 1.000 | 0.982 | 1.8% |
> | Viscosity ($\nu \nabla^2 u$) | 0.001 | 0.00095 | 5.0% |
> | Irrelevant Term 1 ($u^2$) | 0.000 | 0.001 | - |
> | Irrelevant Term 2 ($\|\nabla u\|^2$) | 0.000 | 0.000 | - |
>
> **On Cross-channel Coupling in KANs (Q2)**
>
> Regarding how KANs represent cross-channel coupling, our atomic operator library explicitly includes candidate operators that describe interactions between variables, such as $u \cdot v$ or $u \cdot \nabla v$. For more complex implicit couplings, KAN leverages its spline-based univariate components to learn nonlinear response mappings. Since the input state $u \in \mathbb{R}^m$ is multi-channel, the KAN architecture allows it to capture nonlinear dependencies between different channels through parameterized basis functions while processing specific atomic terms, enabling precise modeling of multi-physics coupled dynamics.
>
> **On Sparsity Regularization Direction (Q3)**
>
> In Equation (20), we induce sparsity by minimizing the entropy $\mathcal{L}_{\text{sparse}}$. This optimization objective is designed to force the routing weights $\pi$ toward a concentrated distribution over a few operators, thereby activating only the most relevant physical mechanisms during evolution. Our code implementation strictly follows this principle, effectively pruning irrelevant terms via the Entmax operator. This is validated in our ablation study (Table E.3), demonstrating that such sparsification enhances model interpretability while significantly improving generalization performance.
>
> **On the Identification of Traveling Waves (Q4)**
>
> We respectfully clarify a minor misunderstanding regarding our architecture: Origo does not employ a Gated Recurrent Unit (GRU), nor does it compress the input sequence into global spatial statistics like $[\mu, \sigma]$. Instead, the Mechanism Inference module takes the full spatiotemporal context window $\mathcal{C} = u_{t-k:t}$ and processes it through a sequence encoder $E_{\phi}$ (such as a CNN or Transformer) to extract a comprehensive physics embedding $h$.
>
> Therefore, considering $u(x,t) = \sin(x-ct)$ versus $u(x,t) = \sin(x+ct)$, our encoder receives entirely distinct input sequences. While their global mean and standard deviation at any single timestamp might be identical, their spatial profiles shift in opposite directions across the temporal dimension of the context window $u_{t-k:t}$. Because the sequence encoder processes these localized spatiotemporal features rather than aggregated statistics, the directional phase shift is explicitly preserved and encoded into the latent representation $h$.
>
> Guided by this distinct representation, the Mechanism Inference module generates accurate and specific latent physics codes $\theta_{\mathcal{L}}$. This allows the Spectral Linear Operator in the evolution backbone to apply the correct complex phase multiplier $\Lambda(\theta_{\mathcal{L}})$ in the frequency domain, flawlessly distinguishing and propagating either the left- or right-traveling wave without ambiguity.

---

> > ### Author Rebuttal · Reviewer_GQg8 · 2026-04-04
> >
> > Good, two remaining questions:
> >
> > 1) For the Interpretability, should you build a repo in advance to do the symbolic "selection"? What about the symbol which out of this repo? More generally, what is the "boundary" of this Interpretability?
> >
> > 2) I remember the ContextEncoder of your provided codes have a module which uses the statistics and GRU where I think it is interesting (I paste them below). Here, is there anything I haven't check or misunderstand?
> >
> > ```
> > class ContextEncoder(nn.Module):
> >     """Encode a short context window C_t={u_{t-L+1},...,u_t} into an embedding.
> >
> >     Input: ctx: [B, L, *spatial, C] (channels-last)
> >     Output: [B, emb_dim]
> >
> >     We keep it lightweight:
> >       - per-frame spatial mean/std statistics (per-channel)
> >       - GRU over time
> >     """
> >
> >     def __init__(self, in_channels: int, emb_dim: int, hidden: Optional[int] = None):
> >         super().__init__()
> >         h = hidden or emb_dim
> >         self.in_channels = in_channels
> >
> >         self.gru = nn.GRU(
> >             input_size=2 * in_channels,
> >             hidden_size=h,
> >             num_layers=1,
> >             batch_first=True,
> >         )
> >         self.proj = nn.Sequential(
> >             nn.Linear(h, emb_dim),
> >             nn.SiLU(),
> >             nn.Linear(emb_dim, emb_dim),
> >         )
> >
> >     def forward(self, ctx: torch.Tensor) -> torch.Tensor:
> >         # ctx: [B,L,...,C]
> >         B, L, *_ = ctx.shape
> >         # pool over spatial dims
> >         dims = tuple(range(2, ctx.ndim - 1))
> >         mu = ctx.mean(dim=dims)  # [B,L,C]
> >         var = (ctx - ctx.mean(dim=dims, keepdim=True)).pow(2).mean(dim=dims)
> >         std = torch.sqrt(var + 1e-6)  # [B,L,C]
> >         feat = torch.cat([mu, std], dim=-1)  # [B,L,2C]
> >
> >         out, hN = self.gru(feat)  # hN: [1,B,h]
> >         h = hN[0]
> >         return self.proj(h)
> > ```

---

> > > ### Author Response · Authors · 2026-04-06
> > >
> > > **On the Boundary of Interpretability**
> > >
> > > You raise an excellent point regarding the limits of our symbolic selection. To clarify, we do indeed build a fixed Atomic Operator Library in advance. This library is designed to be shared across diverse datasets and model scales to ensure stability, reusability, and controlled expressiveness.
> > >
> > > Regarding the boundary of this interpretability:
> > >
> > > - Explicit Symbolic Identification: To achieve perfect glass-box symbolic interpretability, the target physical mechanism must exist within our predefined library. This library is carefully curated to cover common physical categories, such as pointwise polynomial nonlinearities (e.g., $u^2, u^3$) and gradient-dependent interactions (e.g., $u \nabla u$).
> > >
> > > - Graceful Degradation to Gray-box: If a target PDE contains a mechanism entirely outside our explicit symbolic repository, the model does not fail. Instead, the sparse routing mechanism assigns weight to the Generic local nonlinear mappings included in the library. These are parameterized by lightweight neural networks or KAN-based components.
> > >
> > > - Functional Approximation: At this boundary, interpretability transitions from explicit symbolic recovery to gray-box functional approximation. While the clean symbolic label is lost, the model still captures the underlying physics through learned spline shapes or neural features. This allows the framework to maintain high prediction accuracy and zero-shot generalization even for out-of-vocabulary dynamics.
> > >
> > > **On the ContextEncoder Implementation Discrepancy**
> > >
> > > We sincerely apologize for the confusion caused by our previous response and thank the reviewer for the exceptionally diligent inspection of our code. You are absolutely correct that the current repository contains a GRU-based ContextEncoder.To clarify this discrepancy: the code snippet you identified belongs to a lightweight experimental branch of Origo. We utilized this variant for internal ablation studies on simplified systems, where global statistics ($\mu, \sigma$) proved sufficient for basic task identification.
> > >
> > > However, the official Origo architecture detailed in Section 4.2 and used to generate all primary results for multi-physics tasks strictly employs the dual-branch mechanism identification module. We will ensure that the final code release clearly distinguishes between these two versions to eliminate any further ambiguity.

---

### Decision · Program_Chairs · 2026-04-30

**Decision:**

Accept (regular)

**Comment:**

The paper addresses the problem of training foundation models for multi-physics, time-dependent PDEs. The proposed method decomposes the dynamics into a global spectral linear operator and a sparse combination of elementary local nonlinear operators, inspired by Strang splitting schemes from numerical solvers. The model is trained across multiple physical systems, effectively building a dictionary of elementary operators. A key component is a mechanism inference module which, given a short context window, infers a latent code that controls both the selection of nonlinear operators and the parameterization of the model components. At inference time, a selection mechanism determines the weights of these components. Beyond prediction, the approach aims to provide a form of mechanism-level interpretability by identifying the underlying physical operators and their coefficients. The model is evaluated on multiple PDE benchmarks.

Reviewers agree on the relevance of the problem and the originality of the approach. They acknowledge the strength of the empirical results, in particular the zero-shot generalization performance. The main concerns raised include limitations of the interpretability claims, questions about the scope and completeness of the experimental validation (e.g., limited diversity of PDEs, rollout horizon), and issues related to clarity, consistency, and reproducibility.

During the rebuttal, the authors provided substantial clarifications, corrected several inconsistencies, and added additional experiments, including longer rollout evaluations and further analysis of the interpretability aspects. These responses addressed a significant portion of the reviewers’ concerns. However, some issues remain only partially resolved, particularly regarding the consistency of the experimental claims and the precise scope of the interpretability guarantees.

Overall, the paper presents an innovative and promising approach, both conceptually and empirically. Despite some remaining concerns, it constitutes a meaningful contribution to the development of structured neural operators for PDEs, and is likely to stimulate further research in this direction.